# Recruitment dynamics of ESCRT-III and Vps4 to endosomes and implications for reverse membrane budding

**Manuel Alonso Y Adell**[1†§], **Simona M Migliano**[1†§], **Srigokul Upadhyayula**[2,3,4†§], **Yury S Bykov**[5], **Simon Sprenger**[1], **Mehrshad Pakdel**[1,6], **Georg F Vogel**[1,7], **Gloria Jih**[4], **Wesley Skillern**[3], **Reza Behrouzi**[4], **Markus Babst**[8,9], **Oliver Schmidt**[1], **Michael W Hess**[7], **John AG Briggs**[5,10], **Tomas Kirchhausen**[2,3,4‡*], **David Teis**[1,11‡*]

[1]Division of Cell Biology, Biocenter, Medical University of Innsbruck, Innsbruck, Austria; [2]Department of Pediatrics, Harvard Medical School, Boston, United States; [3]Program in Cellular and Molecular Medicine, Boston Children's Hospital, Boston, United States; [4]Department of Cell Biology, Harvard Medical School, Boston, United States; [5]Structural and Computational Unit, European Molecular Biology Laboratory, Heidelberg, Germany; [6]Max Planck Institute of Biochemistry, Martinsried, Germany; [7]Division of Histology and Embryology, Medical University of Innsbruck, Innsbruck, Austria; [8]Department of Biology, University of Utah, Utah, United States; [9]Center for Cell and Genome Science, University of Utah, Utah, United States; [10]Cell Biology and Biophysics Unit, European Molecular Biology Laboratory, Heidelberg, Germany; [11]Austrian Drug Screening Institute, Innsbruck, Austria

**\*For correspondence:**
kirchhau@crystal.harvard.edu (TK);
david.teis@i-med.ac.at (DT)

[†]These authors contributed equally to this work
[‡]These authors also contributed equally to this work
[§]The first three authors are listed in alphabetical order

**Competing interests:** The authors declare that no competing interests exist.

**Abstract** The ESCRT machinery mediates reverse membrane scission. By quantitative fluorescence lattice light-sheet microscopy, we have shown that ESCRT-III subunits polymerize rapidly on yeast endosomes, together with the recruitment of at least two Vps4 hexamers. During their 3–45 s lifetimes, the ESCRT-III assemblies accumulated 75–200 Snf7 and 15–50 Vps24 molecules. Productive budding events required at least two additional Vps4 hexamers. Membrane budding was associated with continuous, stochastic exchange of Vps4 and ESCRT-III components, rather than steady growth of fixed assemblies, and depended on Vps4 ATPase activity. An all-or-none step led to final release of ESCRT-III and Vps4. Tomographic electron microscopy demonstrated that acute disruption of Vps4 recruitment stalled membrane budding. We propose a model in which multiple Vps4 hexamers (four or more) draw together several ESCRT-III filaments. This process induces cargo crowding and inward membrane buckling, followed by constriction of the nascent bud neck and ultimately ILV generation by vesicle fission.
DOI: https://doi.org/10.7554/eLife.31652.001

## Introduction

The ESCRT (endosomal sorting complexes required for transport) machinery mediates 'reverse' membrane budding in distinct but topologically related processes, including formation of intraluminal vesicles (ILVs) in multivesicular bodies (MVBs), plasma-membrane repair and microvesicle formation, abscission in cytokinesis, budding of certain viruses, and nuclear membrane resealing (*Christ et al., 2017*) as well as membrane deformation towards the cytosol (*McCullough et al., 2015*).

MVB formation depends on the hetero-multimers ESCRT-0, ESCRT-I and ESCRT-II, which capture ubiquitinated cargo on the endosomal limiting membrane (*Bilodeau et al., 2002*; *Katzmann et al., 2001*; *Alam et al., 2004*). Budding then requires recruitment of ESCRT-III heteropolymers and the type I AAA+ (ATPase associated with a variety of cellular activities) ATPase, Vps4 (*Babst et al., 2002*). In yeast cells, ESCRT-III includes a major structural component, Snf7, core subunits Vps2, Vps20, and Vps24, and accessory proteins Did2, Vps60 and Ist1 (*McCullough et al., 2013*). Interaction of Vps20 with ESCRT-II triggers membrane recruitment of Snf7, which in turn recruits Vps24 and Vps2. Vps2 recruits Vps4 to these filaments and together with Vps24 may also restrict Snf7 polymerization (*Babst et al., 2002*; *Teis et al., 2008*; *Saksena et al., 2009*; *Teis et al., 2010*). In vitro, Snf7 alone or together with Vps2 and Vps24, forms curved filaments (*Henne et al., 2012*; *Mierzwa et al., 2017*; *Schöneberg et al., 2017*).

In cells, Vps4 forms a hexameric ring when recruited to ESCRT-III filaments (*Caillat et al., 2015*; *Monroe et al., 2014*). A Vps4 hexamer can unfold individual ESCRT-III subunits by ATP-dependent translocation through its central pore (*Scott et al., 2005*; *Yang et al., 2015*; *Monroe et al., 2017*; *Su et al., 2017*); in vitro, this process promotes disassembly of ESCRT-III polymers (*Davies et al., 2010*; *Yang et al., 2015*). The N-terminus of Vps4 contains a 'Microtubule Interacting and Trafficking' (MIT) domain, which interacts with C-terminal 'MIT interaction motifs' (MIM) on ESCRT-III subunits (*Obita et al., 2007*; *Stuchell-Brereton et al., 2007*). Binding of MIT domains to these motifs, exposed in the assembled polymer (*Obita et al., 2007*; *Stuchell-Brereton et al., 2007*), releases Vps4 auto-inhibition (*Merrill and Hanson, 2010*). Formation of ILVs requires coordinated interaction of Vps4 with MIMs of Vps2 and Snf7 (*Adell et al., 2014*).

Current understanding of how the ESCRT machinery promotes reverse membrane budding comes primarily from ensemble biochemical assays with purified components and from live-cell imaging experiments on HIV budding and on plasma-membrane and nuclear-envelop repair and abscission at the midbody using ectopic protein expression. Since ESCRT-III filaments can deform membranes in vivo (*Hanson et al., 2008*) and in vitro (*Wollert et al., 2009*; *Wollert and Hurley, 2010*; *Schöneberg et al., 2017*), it has been suggested that their assembly and/or disassembly provides the driving force for budding (*Fabrikant et al., 2009*; *Chiaruttini et al., 2015*; *Wollert and Hurley, 2010*; *Adell et al., 2014*; *Wemmer et al., 2011*; *Mageswaran et al., 2015*). Live cell imaging of mammalian cells overexpressing fluorescently tagged ESCRT-III and Vps4 showed that ESCRT-III arrived before Vps4 at the sites of HIV budding, during plasma membrane repair and at the nuclear membrane (*Bleck et al., 2014*; *Baumgärtel et al., 2011*; *Prescher et al., 2015*; *Jouvenet et al., 2011*; *Guizetti et al., 2011*; *Vietri et al., 2015*; *Olmos et al., 2015*; *Jimenez et al., 2014*; *Raab et al., 2016*; *Denais et al., 2016*; *Mierzwa et al., 2017*). ESCRT-III and Vps4 are recruited to the midbody prior to abscission, but whether the recruitment is correlated remains to be determined (*Elia et al., 2011*; *Mierzwa et al., 2017*). These experiments did not achieve the required levels of sensitivity, or spatial and temporal resolution to trace the inherently rapid dynamics.

We have now capitalized on the resolution, speed, and noninvasive illumination of lattice light-sheet fluorescence microscopy (LLSM) (*Chen et al., 2014*; *Li et al., 2015*) to study recruitment dynamics of Snf7, Vps24 and Vps4 at individual endosomal carriers imaged over the entire yeast cell volume. During their 3–45 s lifetimes, recruitment of these proteins appeared to be stochastic, but mutually correlated; a large fraction (60%) of the traces did not exceed 62, 33 and 48 molecules (8 hexamers) of Snf7-eGFP, Vps24-eGFP and Vps4-eGFP, respectively. Although non-productive events displayed similar recruitment dynamics, on average they only recruited 12 Vps4 molecules (2 hexamers). Vps4 was not needed to sustain ESCRT-III dynamics, but electron tomography showed that it was essential for ILV budding and scission. To explain how the coordinated activity of ESCRT-III and Vps4 can drive ILV formation, we propose that relatively short ESCRT-III filaments continuously assemble and disassemble. Rapid Vps4 hexamer formation, nucleated by ESCRT-III components, then bridges subunits from two or more ESCRT-III filaments, so that its ATP-dependent unfolding activity can pull (and potentially unfold) an ESCRT-III subunit threaded into its pore while remaining docked on another filament through its MIT domain. At early stages of vesicle formation, uncoordinated ATP-dependent activity of this kind promotes membrane bulging (ILV budding); at later stages, it facilitates membrane fission, release of ILVs and creation of MVBs.

## Results

### Fluorophore tagged Vps4 and ESCRT-III subunits as probes of ESCRT activity on endosomes

We generated fluorophore tagged Snf7 and Vps24 subunits of ESCRT-III and Vps4, to track the dynamics of ILV biogenesis and MVB formation. Direct fusion of eGFP to the C-terminus of ESCRT-III subunits interferes with function (*Teis et al., 2008*), probably because of incompatibility with the periodicity of ESCRT-III polymers (*Tang et al., 2015*) (*Shen et al., 2014*). We therefore inserted the 70-residue localization and affinity purification (LAP) linker between the C-terminus of Snf7 or Vps24 and the N-terminus of eGFP (*Guizetti et al., 2011*). The expression levels of the fusion proteins were comparable to the untagged proteins (*Figure 1—figure supplement 1a*). Direct fusion of eGFP to the C-terminus of Vps4 likewise affects function, because of the positions of the C-terminal helix in higher-order Vps4 assemblies (*Scott et al., 2005*) (*Landsberg et al., 2009*; *Caillat et al., 2015*). We added a 3xHA linker and integrated 3xHA-eGFP or 3xHA-mCherry at the C-terminus of endogenous Vps4 by homologous recombination. Subcellular fractionation showed that Vps4-3xHA-eGFP (Vps4-eGFP) was expressed as a full-length fusion protein at endogenous levels (*Figure 1—figure supplement 1b,c*).

Live-cell, wide-field fluorescence microscopy showed that expression of Snf7-eGFP alone blocked MVB sorting to the vacuole (*Figure 1—figure supplement 1d*), but not when it was co-expressed with endogenous Snf7 (*Figure 1a*). Inspection of the structure of Snf7-polymers (*Tang et al., 2015*; *Shen et al., 2014*) suggests that Snf7-eGFP protomers need to be interspaced with at least 2 untagged Snf7 molecules to avoid steric clashes within the polymer. Vps24-eGFP, Vps4-eGFP and Vps4-mCherry could, however, fully replace the corresponding untagged proteins without compromising trafficking of mCherry-Cps1 or Mup1-mCherry (*Figure 1a*, *Figure 1—figure supplement 1d*). Moreover, wild-type cells expressing Snf7-eGFP along with untagged Snf7 could grow on canavanine, as could cells expressing Vps24-eGFP, Vps4-eGFP as expected for a fully functional MVB pathway (*Lin et al., 2008*), while the deletion mutants of any of these three proteins could not (*Figure 1—figure supplement 1e*). Thus, we have shown that cells tolerate full replacement of Vps4 and Vps24 by their fluorescently tagged equivalents as well as expression of Snf7-eGFP co-expressed with its endogenous, untagged counterpart. Previous work has shown a ratio of Snf7 to Vps24 of about 3–6 in the membrane-bound fraction of wild-type cells (*Teis et al., 2008*). The mean ratio of Snf7-eGFP to Vps24-eGFP measured by calibrated fluorescence intensity in the events studied here is about 1.4:1. In the text below, to estimate the number of total Snf7 molecules in any event, we have therefore multiplied by a factor of 3 the number of eGFP molecules determined directly from the intensity data.

Live-cell, wide-field fluorescence microscopy showed that Snf7-eGFP, Vps24-eGFP and Vps4-eGFP produced both intracellular puncta and diffuse cytoplasmic signals. The puncta fell into two classes: small, mobile, peripheral objects and larger, more static, primarily perivacuolar ones (*Figure 1a*). Vps4-eGFP puncta colocalized with internalized Mup1 on endosomes (*Figure 1—figure supplements 1d* and *2*) and Vps4-mCherry colocalized with Snf7-eGFP and Vps24-eGFP (*Figure 3c, d*). Formation of puncta required the 'early ESCRTs'. Deletion of ESCRT-0 (Vps27), -I (Vps23), or -II (Vps36 or Vps22) left only a diffuse, cytosolic signal for ESCRT-III (Snf7-eGFP or Vps24-eGFP) or Vps4-eGFP (*Figure 1—figure supplement 1f*). Recruitment of Vps24-eGFP depended on Snf7, which in turn required Vps20, and deletion of any ESCRT-III core subunit (Vps20, Snf7, Vps24, Vps2) prevented accumulation of Vps4-eGFP in puncta (*Figure 1—figure supplement 1f*). From these observations, we conclude that the fluorophore tagged Snf7, Vps24, and Vps4 are suitable probes of ESCRT-III activity on endosomes in yeast cells.

### Tomographic analysis of Vps4 containing perivacuolar MVBs

We used high precision in-resin correlative light and electron microscopy (CLEM) (*Kukulski et al., 2011*) to confirm that expression of fluorophore tagged Vps4 did not affect the biogenesis of ILVs and to characterize the morphology and content of the perivacuolar MVBs containing Vps4-eGFP (*Figure 2a–d*) by tomographic reconstruction and volume segmentation (*Figure 2e*). We subjected cells expressing Vps4 fused to a fluorophore – either Vps4-eGFP or Vps4-mNeonGreen (fully functional, *Figure 1—figure supplement 1d*) (*Shaner et al., 2013*) – to high-pressure freezing, freeze-

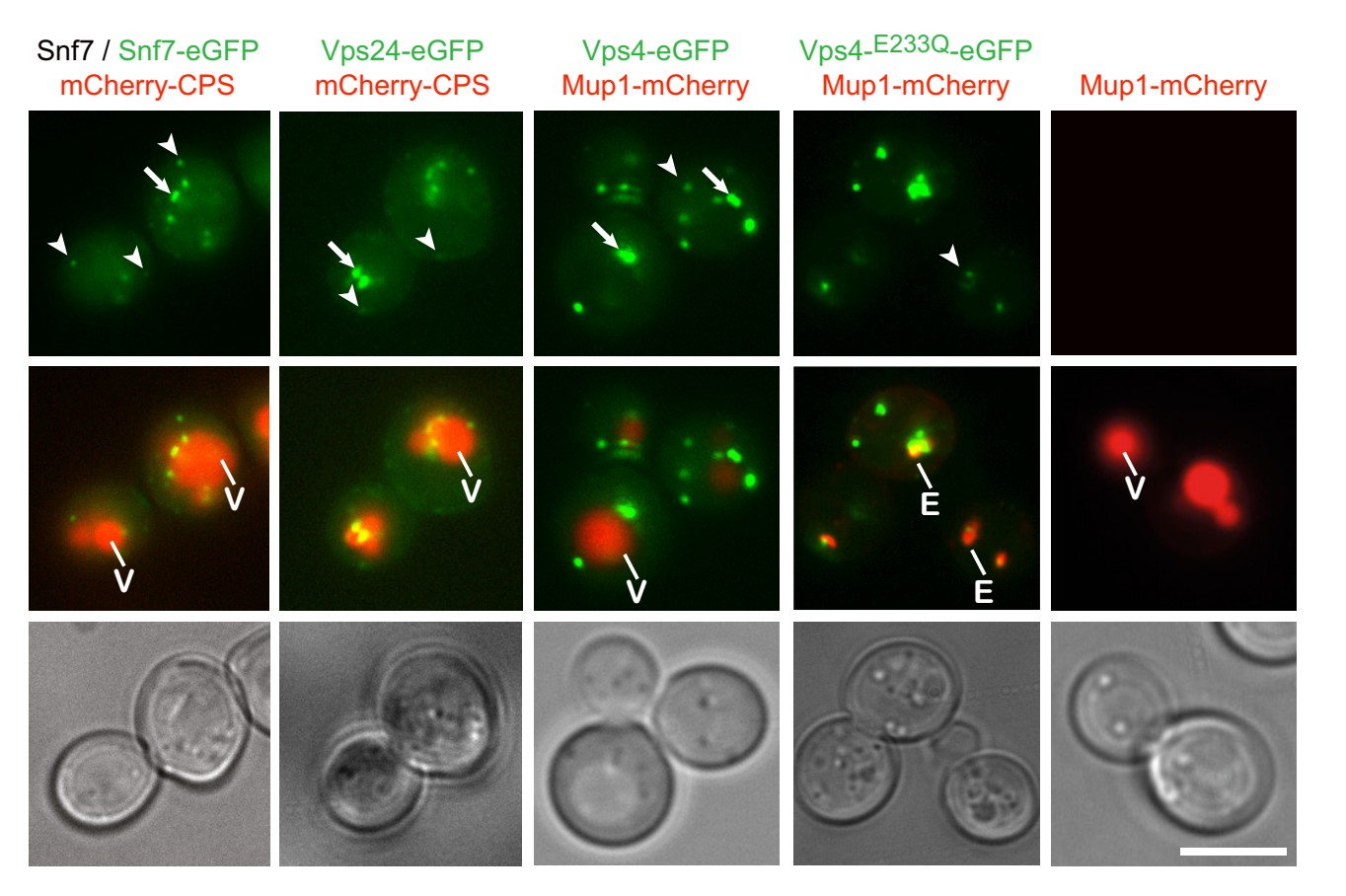

**Figure 1.** Fluorophore tagged ESCRT-III subunits and Vps4 do not interfere with endosomal traffic to the vacuole. Epifluorescence and phase contrast microscopy of live yeast cells expressing Snf7-eGFP mixed with untagged Snf7, Vps24-eGFP, Vps4-eGFP, Vps4$^{E233Q}$-eGFP or control cells, together with fluorescently tagged MVB cargo (mCherry-CPS or Mup1-mCherry), imaged 60 min after methionine addition. Arrowheads and arrows point to small peripheral and to larger perivacuolar objects, respectively; vacuole (V) and class E compartment (E) (red) are indicated. Scale bar = 5 μm.
DOI: https://doi.org/10.7554/eLife.31652.002

The following figure supplements are available for figure 1:

**Figure supplement 1.** Effect of fluorophore tagging of ESCRT-III subunits and Vps4 on the endosomal traffic of cargo to the vacuole.
DOI: https://doi.org/10.7554/eLife.31652.003

**Figure supplement 2.** Internalized cargo traffics through Vps4-containing carriers.
DOI: https://doi.org/10.7554/eLife.31652.004

substitution and sectioning. Immediately after sectioning, we examined the slices on the EM grids by wide field fluorescence microscopy to map the positions of the Vps4-fluorophore signals. We detected 187 Vps4-fluorophore puncta in 1390 sectioned cells, corresponding to approximately two per cell (0.14 fluorescent spots/cell/300 nm-thick section or 1–3 spots/5 μm cell). In living cells, we detected 5–10 Vps4 puncta/cell, indicating loss of signal during the freeze-substitution procedure.

We carried out tomographic analysis on thick sections from 31 cells with 38 Vps4-fluorescent puncta and of four unlabeled cells and found that the 38 puncta correlated with 36 areas with one to three MVBs (a total of 83 MVBs altogether, *Supplementary file 1*). Eleven additional MVB areas (containing 14 MVBs) did not correlate with a fluorescent signal. Because signal preservation was only 20%, we could not distinguish whether the Vps4 signal was low and had been lost during sample preparation or whether these MVBs in fact contained no Vps4. Nevertheless, most of the MVBs (89%) were associated with vacuoles (*Figure 2* and *Supplementary file 1*); a few were associated with other organelles. MVBs contained on average 21 and up to about 65 ILVs of uniform size. The morphologies of Vps4-eGFP or mNeonGreen positive MVBs and of their ILVs were similar to those of cells with untagged Vps4 (*Adell et al., 2014*), further validating the use of Vps4-eGFP expressing

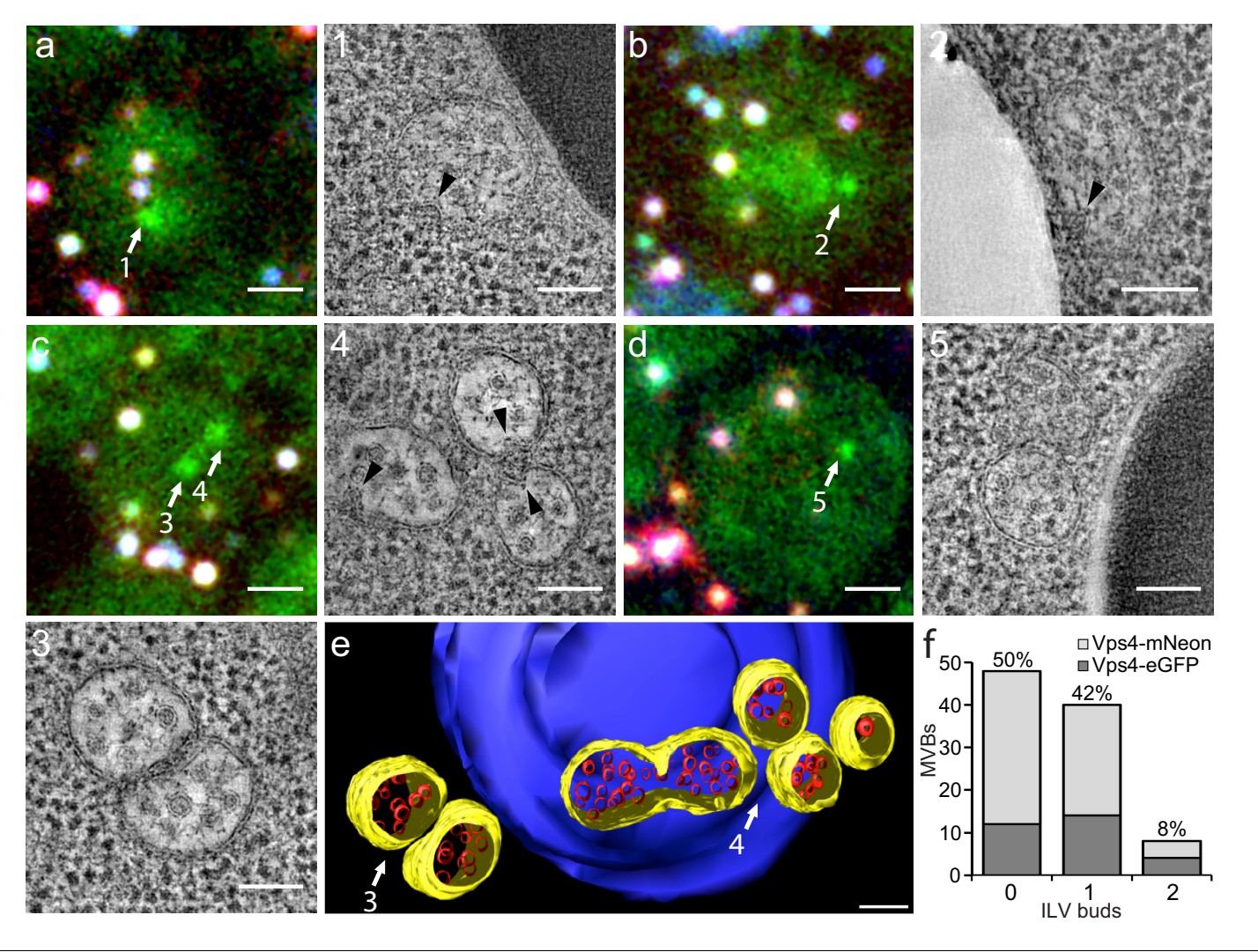

**Figure 2.** Correlative light microscopy and electron tomography of perivacuolar MVBs. Yeast cells expressing Vps4-eGFP or Vps4-mNeonGreen were cryo-fixed and sections subjected to correlative light microscopy and electron tomography. (**a–d**) Representative images from epifluorescence microscopy of 300 nm thick sections, highlighting the presence of Vps4-eGFP perivacuolar fluorescent spots (green). TetraSpeck beads (white) were used to align the light microscopy and electron tomographic images. Scale bar = 1 μm. The highlighted Vps4-eGFP containing objects (1-5) were contained in the sections used to obtain electron tomographic images; the examples illustrate the appearance of endosomes containing ILV (gray spots) and ILV buds (arrowheads). Scale bar = 100 nm. (**e**) 3D model of MVBs imaged by electron tomography corresponding to panel (**c**) showing the MVB limiting membrane (yellow), ILVs (red) and vacuole (blue). Scale bar = 100 nm. (**f**) Histogram distribution indicating the content of ILV buds per MVB determined from the electron tomographic reconstructions from 35 yeast cells with 12 Vps4-eGFP and 26 Vps4-mNeonGreen fluorescent spots. See ***Supplementary file 1*** for details.

DOI: https://doi.org/10.7554/eLife.31652.005

cells for the experiments in this paper. Half of the MVBs (50%) had no ILV buds, 42% had at least one bud and 8% two buds (***Figure 2f***). Up to 30% of possible budding profiles on the top or bottom of a section might have been undetected, due to the missing wedge inherent in electron tomography. Nonetheless, multiple ILV buds appeared with low frequency; we suggest that sequential (rather than concurrent) ESCRT-mediated budding events fill the MVB lumen with ILVs.

## Visualization of ESCRT-III and Vps4 in yeast cells by LLSM

LLSM, a recently introduced visualization tool, enables us to follow molecular events in living cells for relatively long times, with excellent time resolution, and with diffraction limited spatial resolution (***Chen et al., 2014***; ***Kural et al., 2015***; ***Li et al., 2015***; ***Aguet et al., 2016***). A typical 3D time series

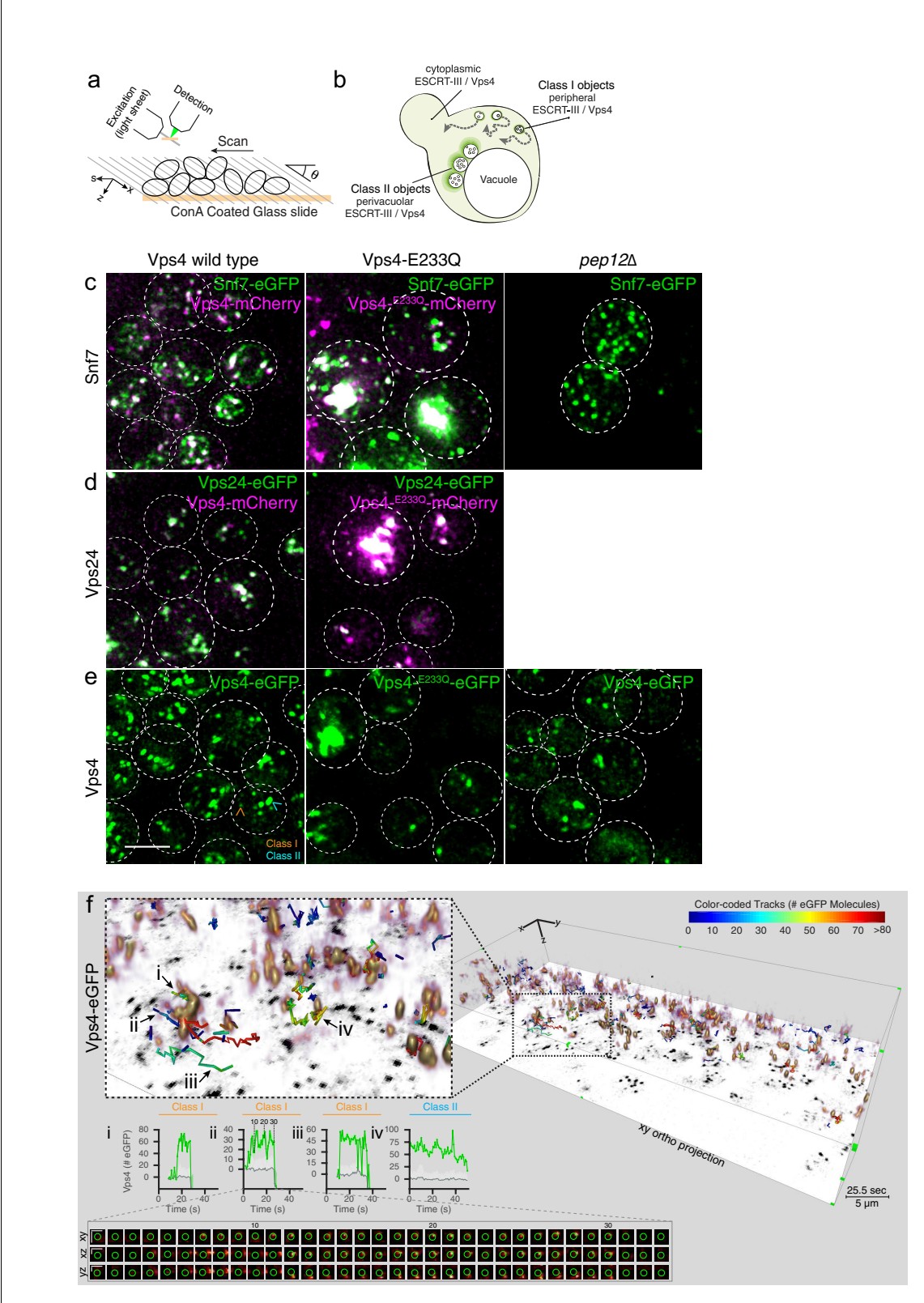

**Figure 3.** 3D Visualization of ESCRT-III and Vps4 recruitment dynamics by LLSM. (a) Schematic representation of the LLSM setup used to obtain time series of 51 s duration from the full cell volume of about 30–50 yeast cells. Image stacks, containing 28–30 sequential imaging optical planes spaced 261 nm apart, were recorded at 21 ms exposure for single or 14.8 ms exposure for alternating excitation wavelengths (b) Schematic model showing the location of fluorescent ESCRT-III or Vps4-eGFP expressed in yeast cells, showing relatively small peripheral mobile spots (class I objects), larger and

*Figure 3 continued on next page*

*Figure 3 continued*

relatively static perivacuolar spots (class II objects), and a diffuse cytosolic signal. (**c–e**) Representative orthogonal views after 3D deconvolution of WT yeast cells and the indicated mutants expressing either a mixture of Snf7 and Snf7-eGFP together with Vps4-mCherry, Vps24-eGFP together with Vps4-mCherry or Vps4-eGFP alone. Examples of class I and class II objects in panel (**e**) are indicated. Scale bar 3 μm. (**f**) Volume rendering of LLSM image from a single time point (25.5 s) obtained from cells expressing Vps4-eGFP (see also *Video 2*). The right-hand panel includes the orthogonal projection along the xy axis (black) and illustrates the complexity of the 3D image. Scale bar = 5 μm. The left-hand panel shows an enlarged region, overlaid with tracks of representative, diffraction-limited fluorescent objects (class I, tracks i-iii) of increasing lifetimes and a long lived non-diffraction limited perivacuolar object (class II, track iv); these were traced automatically in 3D and in time, for their content (color-coded) of fluorescent molecules. Each plot compares the number of fluorescent molecules converted from the fluorescence signal of the spot (green) with the local background (gray) and includes the 95% confidence interval of the measurements (light gray fill). Additional traces are shown in *Figure 4—figure supplements 2–9*. The bottom panel shows a cross sectional time montage of the deconvolved object (orange) corresponding to track ii. Scale bar = 1 μm.

DOI: https://doi.org/10.7554/eLife.31652.013

The following figure supplement is available for figure 3:

**Figure supplement 1.** Visualization of ESCRT-III and Vps4$^{E233Q}$ recruitment dynamics by LLSM.

DOI: https://doi.org/10.7554/eLife.31652.014

was 51 s long and covered about 10–50 cells; each volume was acquired in 750–800 ms with a 100 ms pause between scans (*Videos 1–7*). A full LLSM dataset contained 10–20 3D-time series from 300 to 1000 cells. Within the full volumes of living yeast cells (*Figure 3a*), we tracked the 3D positions of ESCRT-III and Vps4-containing objects (a total of 5,000–12,000 diffraction-limited objects), with 850 – 960 ms time resolution and three eGFP-molecule sensitivity (data for Snf7-eGFP, Vps24-eGFP and Vps4-eGFP are summarized in *Supplementary file 2*).

In WT cells expressing Snf7-eGFP together with endogenous Snf7, Vps24-eGFP or Vps4-eGFP (*Videos 2–4*), we detected two classes of fluorescent signals: class I, with diffraction-limited objects distributed throughout the cytosol, and class II, with a mixture of diffraction-limited and non-diffraction limited fluorescent objects, both of relatively high intensity and low mobility, located close to the vacuole (*Figure 3b*). Typically, a cell contained 5–10 mobile, diffraction-limited puncta (i.e., < 300 × 300 x 600 nm), each relatively weak. The size is compatible with the sizes of individual MVBs in yeast (<250 nm). When co-expressed, Snf7-eGFP and Vps24-eGFP colocalized with Vps4-mCherry (*Figure 3c,d*).

Our analysis of ESCRT-III and Vps4 dynamics focused on the diffraction-limited and mostly peripheral endocytic carriers (class I). The data in *Figure 1—figure supplement 2* demonstrated that they carry endocytic cargo. Correlative electron microscopy, described above, revealed that the larger class II objects are pairs or small clusters of perivacuolar MVBs (*Figure 2e*). LLSM 3D visualization showed that cargo reached them 5–10 min after the onset of endocytosis (*Video 8*, *Figure 1—figure supplement 2*). Although we could not resolve molecular events in these class II objects (*Figure 4—figure supplement 1*) as distinctly as we could with the peripheral class I objects (*Figure 4*, *Figure 4—figure supplements 2–4*), the patterns of intensity variation for Snf7, Vps24 and Vps4, described below, support the interpretation that ESCRT dynamics on these MVBs are similar to the dynamics that we analyzed in detail for the peripheral endosomes (*Figure 4*, *Figure 4—figure supplements 1–4*).

To illustrate the complexity of the 3D data, we present orthogonal (*Figure 3c-e*) and volumetric (*Figure 3f*, *Figure 3—figure supplement 1*, *Videos 2–4*) projections at a single time point in time series recorded from yeast cells expressing, Snf7-eGFP together with Vps4-mCherry (*Figure 3c*), Vps24-eGFP together with Vps4-mCherry (*Figure 3d*), Vps4-eGFP alone (*Figure 3e*). The selected examples in these figures (*Figure 3f*, *Figure 3—figure supplement 1a,b*, *Videos 2–4*) show tracks color-coded by the number of eGFP molecules that accumulated paired with corresponding fluorescence intensity plots.

## Analysis of ESCRT-III and Vps4 dynamics in peripheral endosomes

A common feature of all traces was an abrupt increase from baseline in the fluorescence intensity for ESCRT-III subunits Snf7 or Vps24, to levels of about 60 and 14 subunits, respectively. In most cases, the intensity then fluctuated with no detectable pattern and no net accumulation until an abrupt decline to baseline after a variable time interval (*Figure 4—figure supplements 2* and *3*).

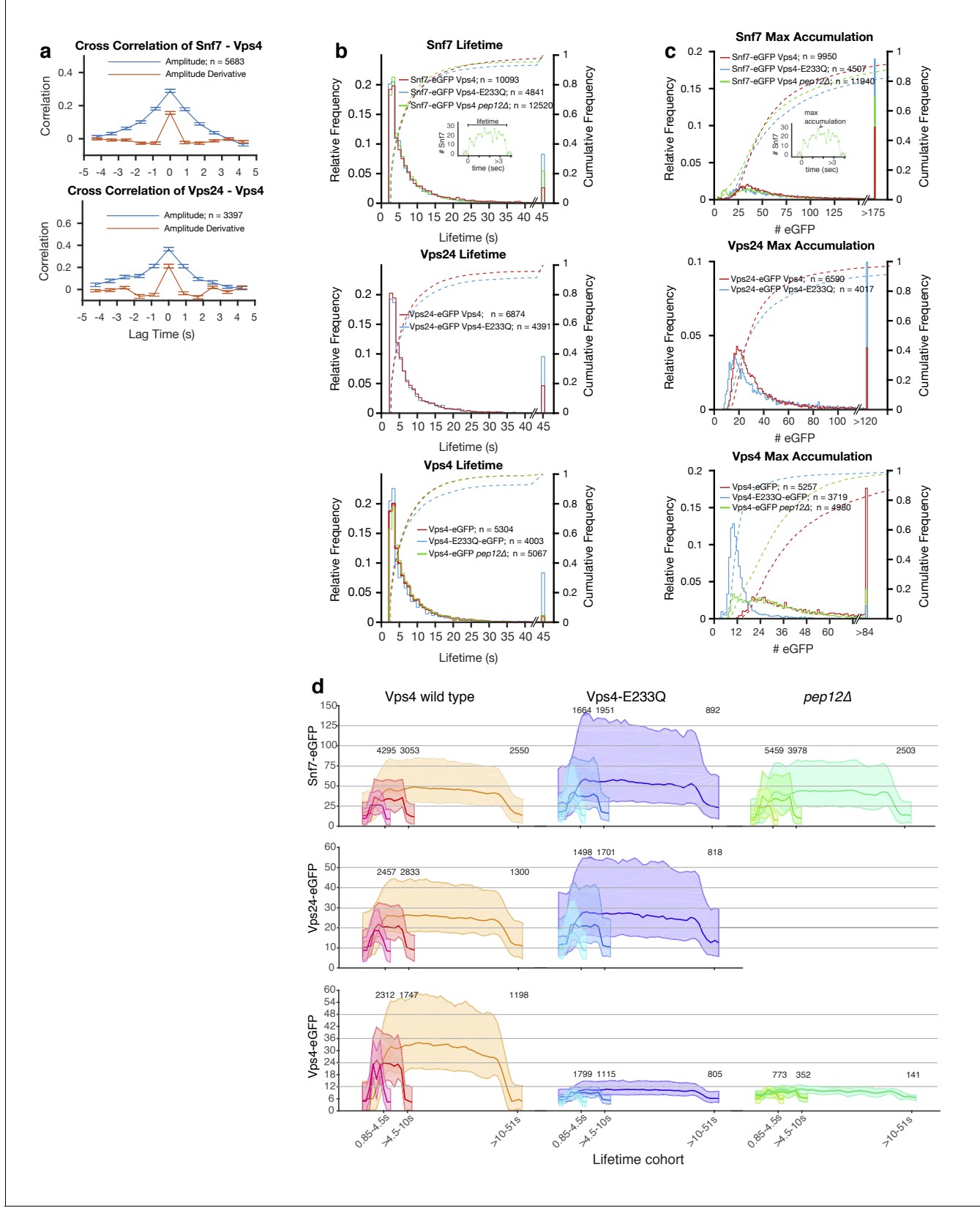

**Figure 4.** Analysis of ESCRT-III and Vps4 recruitment to peripheral endosomes. Quantitative analysis of time series acquired with the LLSM over the full volume of 300–1000 yeast cells expressing a mixture of Snf7 and Snf7-eGFP with either Vps4-mCherry or Vps4[E233Q]-mCherry mutant, Vps24-eGFP with

*Figure 4 continued on next page*

Figure 4 continued

Vps4-mCherry or Vps4$^{E233Q}$-mCherry, and Vps4-eGFP or Vps4$^{E233Q}$-eGFP. Snf7-eGFP and Vps4-eGFP was also analyzed in *pep12Δ* mutants. The data presented are from all diffraction limited mobile objects (class I) detected in the periphery of cells (a, c, d) or in both peripheral and perivacuolar regions (b). (a) Cross-correlation of the fluorescence intensity (blue) and of the fluorescence intensity first derivative (orange) from Snf7-eGFP and Vps4-mCherry or from Vps24-eGFP and Vps4-mCherry. Data are from traces with lifetimes longer than 11 s and are expressed as average ± SD. (b) Plots showing the lifetime distribution (histogram) and corresponding cumulative frequency distribution (dotted curves) of Snf7-eGFP, Vps24-eGFP and Vps4-eGFP in WT cells and in the indicated mutants. The two-sample permutation test for differences between the medians was not significant. The number of tracked traces analyzed for each experiment is indicated. The inset showing a typical trace illustrates the definition of lifetime. (c) Plots showing the maximum accumulation (histogram) and corresponding cumulative frequency (dotted curve) distributions of fluorescent molecules of Snf7-eGFP, Vps24-eGFP and Vps4-eGFP in WT cells in the indicated mutants. Mutating Vps4 had minimal effects on the modes of maximum Snf7-eGFP recruitment (35 ± 12 and 30 ± 10, amplitude ± SD of the first fitted Gaussian, for wild-type and Vps4$^{E233Q}$ mutant, respectively) or of Vps24-eGFP (21 ± 5 and 17 ± 6; p<0.001, Kolmogorov-Smirnov and the two-sample permutation tests). Vps4$^{E233Q}$ or loss of Pep12 had a marked effect on the accumulation of Vps4-eGFP itself (from 24 ± 6 to 11 ± 3 and 12 ± 3 in wild-type Vps4, Vps4$^{E233Q}$, and *pep12Δ* mutants, respectively; p<0.001). The inset of a typical trace illustrates the definition of maximum accumulation. (d) Averaged number of eGFP molecule traces per lifetime cohort, shown as mean ± 95th percentile confidence bound (shaded areas) for all traces above the local background threshold analyzed in (c). The data is for Snf7-eGPF, Vps24-eGFP and Vps4-eGFP expressed in the indicated wild type and mutant yeast cell strains. The Vps4-eGFP data from the *pep12Δ* mutant corresponds to traces likely to be associated with a single endocytic carrier; they correspond to events whose maximum accumulation of Vps4-eGFP molecules were within the 99th percentile of the first Gaussian distribution (**Figure 4—figure supplement 10f**). The complete data set is shown in **Figure 4—figure supplement 10g**.
DOI: https://doi.org/10.7554/eLife.31652.015

The following figure supplements are available for figure 4:

**Figure supplement 1.** Analysis of ESCRT-III and Vps4 recruitment associated with perivacuolar endosomes.
DOI: https://doi.org/10.7554/eLife.31652.016
**Figure supplement 2.** Traces of Snf7-eGFP and Vps4-mCherry obtained by LLSM.
DOI: https://doi.org/10.7554/eLife.31652.017
**Figure supplement 3.** Traces of Vps24-eGFP and Vps4-mCherry obtained by LLSM.
DOI: https://doi.org/10.7554/eLife.31652.018
**Figure supplement 4.** Traces of Vps4-eGFP obtained by LLSM.
DOI: https://doi.org/10.7554/eLife.31652.019
**Figure supplement 5.** Traces of Snf7-eGFP in *pep12Δ* mutants obtained by LLSM.
DOI: https://doi.org/10.7554/eLife.31652.020
**Figure supplement 6.** Traces of Vps4-eGFP in *pep12Δ* mutants obtained by LLSM.
DOI: https://doi.org/10.7554/eLife.31652.021
**Figure supplement 7.** Traces of Snf7-eGFP and Vps4$^{E233Q}$-mCherry obtained by LLSM.
DOI: https://doi.org/10.7554/eLife.31652.022
**Figure supplement 8.** Traces of Vps24-eGFP and Vps4$^{E233Q}$-mCherry obtained by LLSM.
DOI: https://doi.org/10.7554/eLife.31652.023
**Figure supplement 9.** Traces of Vps4$^{E233Q}$-eGFP obtained by LLSM.
DOI: https://doi.org/10.7554/eLife.31652.024
**Figure supplement 10.** Composition of Vps4 in the cytosol of wt cells and on the endosomes in *pep12Δ* mutants.
DOI: https://doi.org/10.7554/eLife.31652.025

*Supplementary file 2* summarizes our analysis from cells that expressed Snf7-eGFP or Vps24-eGFP together with Vps4-mCherry. Although all the Vps4-mCherry traces coincided with Snf7-eGFP or Vps24-eGFP traces (*Figure 3b,c*), only in about 32% of them (more than 3300 traces) was the Vps4-mCherry signal sufficiently high for subsequent analysis. The intensity fluctuations of Snf7-eGFP and of Vps24-eGFP correlated with those of Vps4-mCherry in the same trace (*Figure 4a*). In both cases, the cross-correlation of amplitude and its derivative had single peaks at the origin, indicating coordinated recruitment dynamics of Vps4 and the ESCRT-III components, at least on a time scale longer than the resolution of our determinations (850 ms). Though the fast photobleaching of Vps4-mCherry precluded the correlational analysis of the remaining traces, the lifetime (the interval between the appearance and disappearance of above-baseline intensity) of Snf7-eGFP, Vps24-eGFP and Vps4-eGFP (*Figure 4b,d*) and dynamics properties (*Figure 4*, *Figure 4—figure supplements 2–4*) were indistinguishable.

The lifetime followed similar bi-exponential distributions for all Snf7 and Vps24 events (*Figure 4b*, *Supplementary file 2*). The maximum amount of Snf7-eGFP in the traces had a relatively broad distribution (*Figure 4c*), with modes that ranged from 30 ± 9 during the initial 750 ms to 38 ± 12 for the longest-lived structures (10–51 s) (*Supplementary file 2*). Thus, these structures would contain an

overall average of between 75 and 200 Snf7 subunits, with very few of the structures containing 200 or more (*Figure 4c,d*). We can be more definite about the Vps24 stoichiometry, because we could achieve full replacement of endogenous Vps24 with Vps24-eGFP (*Figure 4c,d*). Vps24 recruitment had a broad range with modes that ranged from 14 ± 3 during the initial 750 ms to 22 ± 6 for the longest-lived structures (10–51 s) (*Supplementary file 2*). Similar distributions were obtained for the mean accumulation of Snf7-eGFP and Vps24-eGFP during the lifetime of the events (*Figure 4—figure supplement 1c*). Assuming that all ESCRT-III assemblies have both subunits, the ratio of Snf7 to Vps24 ranges between three or five to one, in agreement with published biochemical data (*Teis et al., 2008*).

In all events that began and ended within the 51 s duration of the movies, the distribution of lifetimes for Vps4-eGFP, like those for Snf7 and Vps24, was bi-exponential (*Figure 4b*). During the initial 750 ms, 20 ± 5 molecules (~3 hexamers) arrived (*Supplementary file 2*). About 20% of the events never accumulated more than 3 hexamers at any one time (*Figure 4c*) (although they gained and lost Vps4 stochastically throughout their lifetime); most of these events were of short duration. The 80% that acquired 4 or more hexamers at any one time were longer lived (≥5 s) (*Figure 4b–d*). Non-productive events in cells expressing a Vps4 mutant defective in ATP hydrolysis (E233Q) or in cells lacking Pep12 (see below) only rarely accumulated more than 3 Vps4 hexamers (*Figure 4d*, *Supplementary file 2*).

Using a spinning disc confocal microscope calibrated for single-molecule counting (*Cocucci et al., 2012*), we determined that nearly all the cytosolic Vps4-eGFP was monomeric in yeast cells, whether or not the cells were exposed to chemical crosslinking before lysis and adsorption of the cytosol onto a glass coverslip (*Figure 4—figure supplement 10a–d*). These observations suggest that Vps4 hexamers form during recruitment of ESCRT-III to endosomes.

## Analysis of ESCRT-III and Vps4 dynamics in perivacuolar MVBs

On perivacuolar MVBs with persistent fluorescence we detected brighter signals of Snf7-eGFP, Vps24-eGFP and Vps4-eGFP; the maximum accumulations had a relatively broad distribution with a mode 4–5 times larger (*Figure 4—figure supplement 1a*) than in the peripheral objects (*Figure 4c*). This is the behavior expected for optically unresolved clusters of perivacuolar MVBs visualized by electron microscopy (*Figure 2*). We also found that the short-time fluctuations in the numbers of Snf7-eGFP, Vps24-eGFP and Vps4-eGFP molecules (*Figure 4—figure supplement 1b*) corresponded to the numbers of Snf7-eGFP, Vps24-eGFP and Vps4-eGFP molecules in the spatially resolved events detected in diffraction limited peripheral objects (*Figure 4c*). Presumably they represented individual budding events we observed by CLEM (*Figure 2*).

## ESCRT-III and Vps4 dynamics on endosomes lacking intraluminal vesicles

Fusion of the early, small endocytic carriers is a prerequisite for MVB formation. Pep12 (the yeast syntaxin homolog) is essential for fusion; its absence leads to the accumulation of 40–50 nm diameter endosomes, which are too small to accommodate ILVs, and thus fail to deliver endocytic cargo into the vacuole (*Becherer et al., 1996*). Electron micrographs of Pep12 deficient (*pep12Δ*) cells expressing Vps4-eGFP showed eGFP-specific antibody labeling on small vesicles, ~45 ± 18 nm in diameter (n = 439) devoid of ILVs, as expected (*Figure 4—figure supplement 1d*). In most of the images, 2–3 vesicles formed small clusters. As a control for correct localization of the immuno-gold label, we examined cells expressing either Snf7-eGFP or Vps4-eGFP together with Vps21 (Rab5), which clustered the MVBs (*Adell et al., 2014*) and facilitated their detection. In these cells, the gold-labeled, eGFP-specific antibody marking Snf7-eGFP or Vps4-eGFP indeed associated with MVBs (*Figure 4—figure supplement 1d*).

LLSM imaging of *pep12Δ* cells expressing Snf7-eGFP or Vps4-eGFP showed fluorescent puncta, spatially distributed like the peripheral and perivacuolar endosomes seen in WT cells (*Figure 3c,e*). The lifetimes of the peripheral events were similar to those of comparable events in WT cells and cells expressing Vps4$^{E233Q}$ (*Figure 4b*, *Figure 4—figure supplements 5* and *6*, see below). The intensity distribution of Snf7-eGFP and Vps4-eGFP in *pep12Δ* cells had two distinct peaks (*Figure 4c*). The distribution for Vps4-eGFP corresponded to 2 and 4 hexamers, respectively (*Figure 4c*, *Figure 4—figure supplement 10f,g*, *Supplementary file 2*). The former peak likely

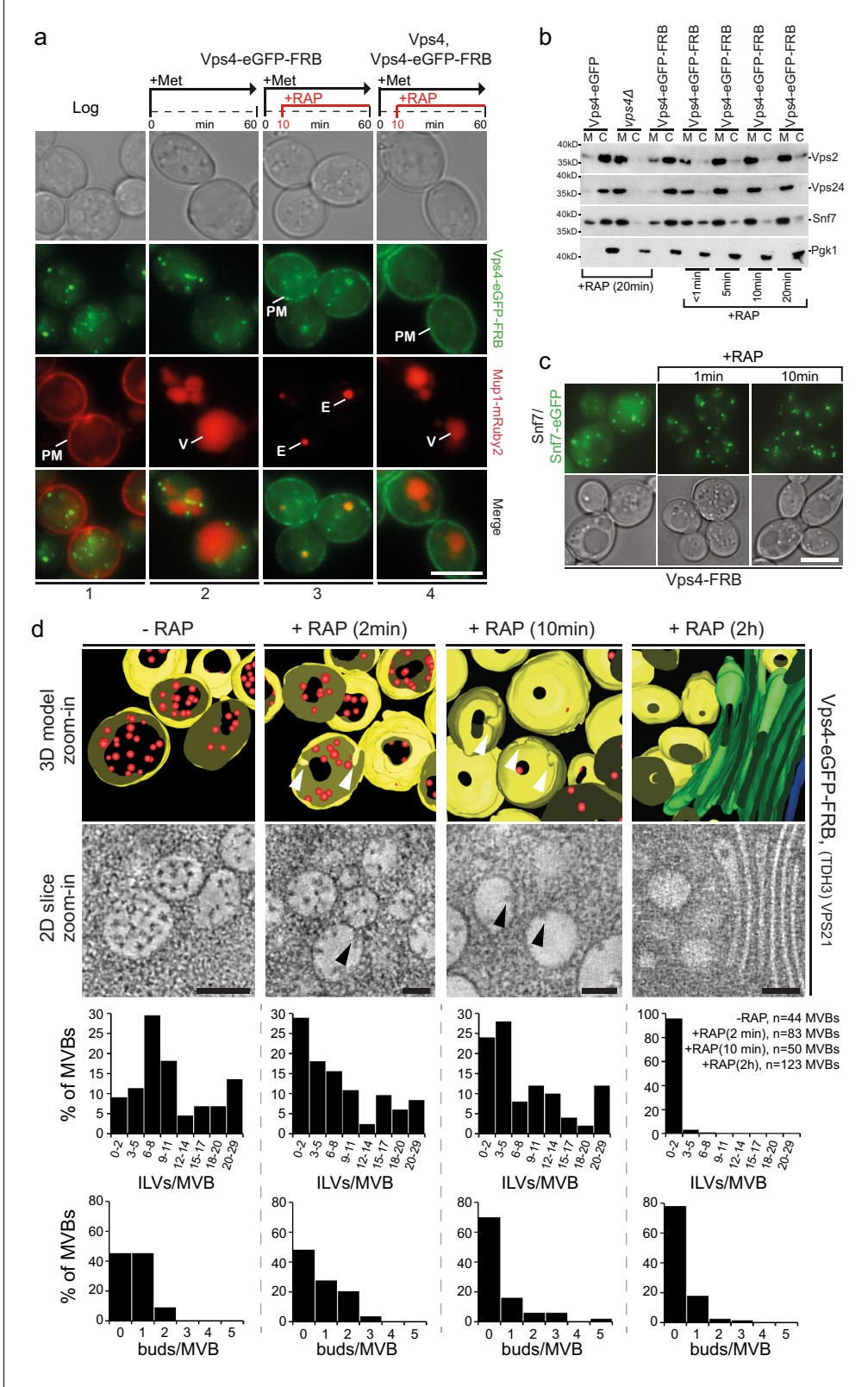

**Figure 5.** Acute cytosolic depletion of Vps4 blocks endosomal traffic to the vacuole and changes the structure of perivacuolar MVBs. Rapid depletion of cytosolic Vps4-eGFP-FRB was achieved by its capture on the cytosolic surface of the plasma membrane induced by rapamycin-mediated heterodimerization with Pma1-FKBP12. (a) Phase contrast and live cell epifluorescence microscopy of yeast cells expressing Vps4-eGFP-FRB (green) and the cargo Mup1-mRuby2 (red) before (panel 1) or 60 min after addition of methionine (panels 2–4), used to activate endocytosis of Mup1 and its

*Figure 5 continued*

transport into the vacuole. Rapamycin was added to deplete Vps4-eGFP-FRB (panel 3, 4), and untagged Vps4 was co-expressed to rescue the anchor-away effect (panel 4). The locations of the plasma membrane (PM), vacuole (V) and class E compartment (E) are indicated. Scale bar = 5 µm. (b) SDS-PAGE and western blot analysis with the indicated antibodies of samples from equal volumes of membrane and cytosolic fractions of Vps4-eGFP cells, *vps4Δ* mutants and Vps4-eGFP-FRB cells in the absence and presence of rapamycin. (c) Live cell epifluorescence and phase contrast microscopy of yeast cells expressing a mixture of Snf7 and Snf7-eGFP together with Vps4-FRB before or after the addition of rapamycin. Scale bar = 5 µm. (d) 3D model (top) and 2D slices (bottom) of tomographic reconstructions obtained from 400 nm slices of yeast cells before and after addition of rapamycin. The images show typical examples of MVBs containing ILVs (red) surrounded by the limiting membrane (yellow), endosomes lacking ILVs and an example of class E-like compartment (green); arrowheads point to ILV buds. To facilitate the imaging, MBVs were clustered in one region by overexpression of Vps21 (TDH3-*VPS21*) (*Adell et al., 2014*). Scale bar = 100 nm. The outcome of the morphological analysis of all MVBs (fully or only partially present in the section) is shown.

DOI: https://doi.org/10.7554/eLife.31652.027

The following figure supplement is available for figure 5:

**Figure supplement 1.** Effects of the acute depletion of the cytosolic Vps4 pool.
DOI: https://doi.org/10.7554/eLife.31652.028

represented single small endocytic carriers while the latter likely represented the clusters of at least two small endocytic carriers seen in the EM images. The dynamics of Snf7 was similar to WT cells (*Figure 4d*). In contrast, the Vps4 dynamics at peripheral events in *pep12Δ* cells (*Figure 4d*) resembled those in Vps4$^{E233Q}$ cells more closely than they did those in WT cells.

## Role of Vps4 ATP hydrolysis in ESCRT-III recruitment dynamics

Constitutive expression of Vps4$^{E233Q}$, a mutant that is impaired in ATP hydrolysis, causes disappearance of MVBs and formation instead of flat, cisternal membrane stacks - the 'class E compartment' - on which ESCRT-III complexes and Vps4 accumulate (*Babst et al., 1997*; *Babst et al., 1998*). LLSM imaging of cells expressing Vps4$^{E233Q}$-eGFP as the only Vps4 species showed a substantial entrapment of Vps4$^{E233Q}$-eGFP (*Figure 3e*, *Figure 3—figure supplement 1e*, *Video 5*) and ESCRT-III subunits in class E compartments (*Figure 3c,d*, *Figure 3—figure supplement 1c,d*, *Videos 6* and *7*). These large, very bright, relatively static perivacuolar objects always had lifetimes longer than the time series. We also detected in these cells diffraction limited, mobile, peripheral objects (about 5–10 detected per yeast cell within the 51 s time window) (*Figure 3c–e*, *Videos 5–7*). The distribution of lifetimes for 93% of the peripheral objects in cells expressing Vps4$^{E233Q}$-eGFP was almost identical to the distribution of similar objects in cells only expressing wild-type Vps4-eGFP (*Figure 4b*, *Figure 4—figure supplement 9*, *Supplementary file 2*). The remaining peripheral objects persisted beyond the 51 s time window suggesting impaired or slower release of Vps4$^{E233Q}$-eGFP unable to hydrolyze ATP (*Figure 4b*, *Supplementary file 2*).

We conclude from this unexpected result that Vps4 lifetimes in peripheral endosomes are independent of Vps4 ATPase activity and that association and release of Vps4 can occur in the absence of ATP hydrolysis. We confirmed this conclusion by analyzing the recruitment patterns of Snf7-eGFP and Vps24-eGFP in cells expressing Vps4$^{E233Q}$-mCherry. We found Snf7-eGFP and Vps24-eGFP in both static, relatively bright, long lived perivacuolar objects (most likely class E compartments) and mobile diffraction-limited objects in the peripheral cytosol (*Figure 3c,e*, *Figure 3—figure supplement 1c,d*, *Videos 6* and *7*). Recruitment of Snf7-eGFP or of Vps24-eGFP to the peripheral objects, in cells expressing Vps4$^{E233Q}$-mCherry, had essentially the same dynamic characteristics as in cells expressing wt Vps4-mCherry (*Figure 4c* - d, *Figure 4—figure supplements 7* and *8*; *Supplementary file 2*). Expression of Vps4$^{E233Q}$-mCherry led to the accumulation of Snf7-eGFP in a fraction of peripheral and all perivacuolar endosomes (*Figure 4c,d*, *Figure 4—figure supplement 1a–c*). Thus ESCRT-III assembly and release events occurred on endosomes but absence of Vps4 ATPase activity means that none result in ILV budding.

We did detect important differences between traces in mutant and wild-type Vps4-eGFP expressing cells. The intensity distribution of the peripheral, diffraction limited Vps4$^{E233Q}$-eGFP containing structures peaked at about 11 ± 3 molecules, that is, about two Vps4 hexamer – less than the maximum of 4 or more found with the wild-type protein (*Figure 4c*, *Figure 4—figure supplement 10f*; *Supplementary file 2*). Moreover, the intensity distribution for the mutant was independent of the

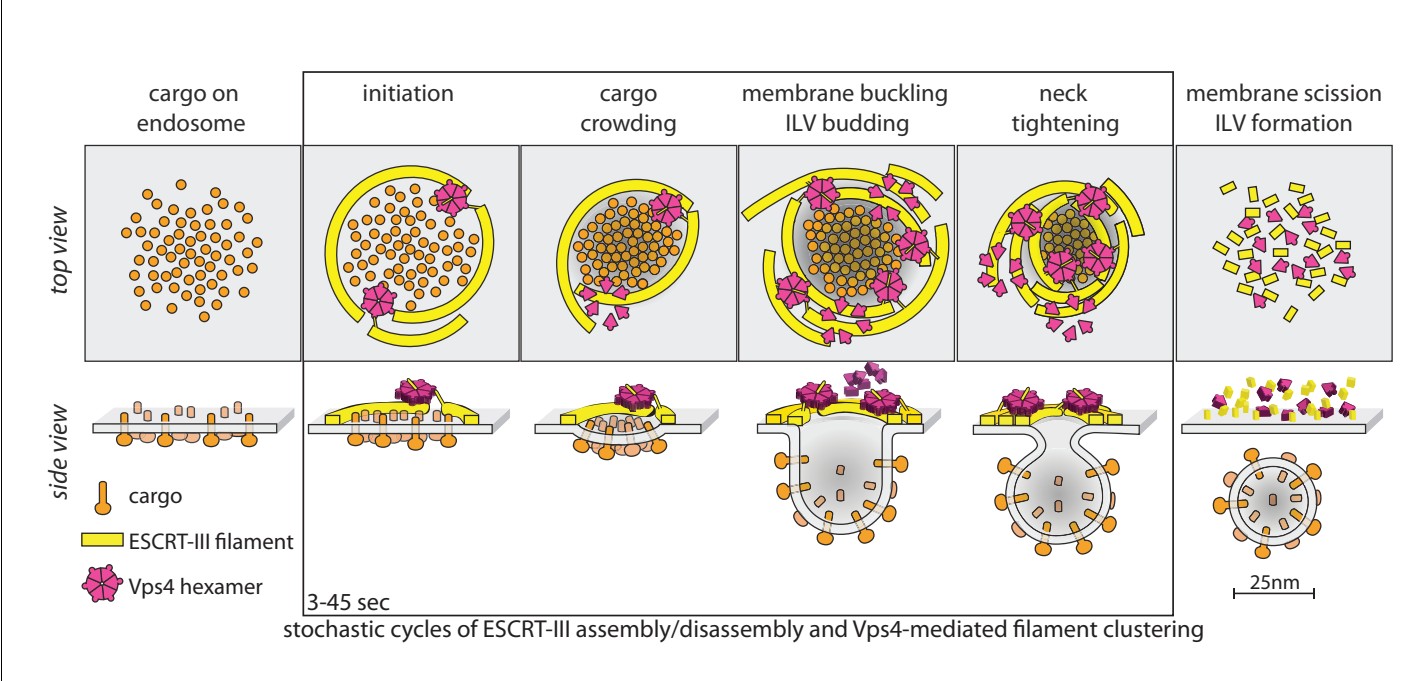

**Figure 6.** Proposal of a model for the mechanism of ESCRT-mediated intraluminal vesicle formation. The figure shows a schematic representation for possible stages during the budding and scission of ILVs in MVBs. ESCRT-0-II (not shown) bind cargo proteins and nucleate rapid assembly of several ESCRT-III filaments. These ESCRT-III assemblies grow and shrink stochastically and recruit Vps4. ESCRT-III dynamics occur continuously throughout the formation of ILVs and do not require Vps4. When recruited, Vps4 hexamers, docked by MIT-MIM association on one filament, exerts ATP-dependent force on another filament by pulling the C-terminus of an ESCRT-III subunit into the hexamer pore; as the cycling process ensues, the filament movement drives crowding of the membrane-anchored cargo. The crowding forces membrane invagination, to create the surface area needed to accommodate the cargo molecules. The forces exerted by the interactions between ESCRT-III and Vps4 hexamers also generate the neck of the nascent bud, lead to membrane constriction and ultimately to vesicle fission and final release of the ESCRT machinery.

DOI: https://doi.org/10.7554/eLife.31652.029

lifetime of the object, unlike wild-type Vps4-eGFP, for which longer traces were in general brighter (*Figure 4d*).

The levels of Vps4$^{E233Q}$-eGFP at peripheral endosomes are similar to those of Vps4-eGFP in *pep12Δ* cells (*Figure 4d*). Both mutations fail to support ILV formation - in Vps4$^{E233Q}$-eGFP cells because there is no source of energy for deforming the membrane and in *pep12Δ* cells because the unfused carrier vesicles are too small.

## Effects of acute cytosolic depletion of Vps4

We used the 'anchor away' method (*Haruki et al., 2008*) to eliminate Vps4 activity by acutely reducing available cytosolic Vps4 without the confounding effects of long-term expression of a non-functional mutant (*Figure 5—figure supplement 1a*). Rapamycin-induced heterodimerization of fully functional Vps4-eGFP-FRB (FRB: FKBP12 rapamycin binding domain) (*Figure 5a*), with Pma1-FKBP12 (the abundant plasma-membrane H$^+$-ATPase, Pma1, fused to the 12 kDa FK506 binding protein, FKBP12) rapidly depleted the cytoplasmic Vps4-eGFP-FRB pool. Within a minute after rapamycin addition, most of the Vps4-eGFP-FRB had anchored irreversibly to Pma1-FKBP12 (*Figure 5—figure supplement 1b*).

The consequences of acute Vps4 depletion (*Figure 5b,c*) were similar to long-term expression of Vps4$^{E233Q}$ or of Vps4 deletion (*Figure 1—figure supplement 1b,c*), causing decreased cytosolic ESCRT-III. Some Vps4 together with ESCRT-III even remained on perivacuolar endosomes, as if trapped on a class E compartment (*Figure 5—figure supplement 1c*). FRAP experiments (*Figure 5—figure supplement 1c*) confirmed that after cytosolic pool depletion, photobleached Vps4-eGFP-FRB puncta barely recovered and that no new Vps4 could be recruited from the cytosol.

Live cell microscopy showed that depletion of Vps4 prevented delivery of cargo to the vacuole and caused instead an accumulation of MVB cargo molecules in perivacuolar endosomes (*Figure 5a*). Electron tomography showed directly the consequences of acute Vps4 depletion on ILVs and on the morphology of perivacuolar structures (*Figure 5d*, *Figure 5—figure supplement 1d*). We subjected cells to high-pressure freezing 2, 10 and 120 min after adding rapamycin. Although ESCRT-III was still recruited, the number of ILVs per MVB decreased, while the number of MVBs with ILV buds did not increase (*Figure 5d*). In contrast, after 10 min the number of endosomes lacking ILV buds increased (*Figure 5d*). These observations suggest that ILV budding (e.g. membrane buckling) and scission stall when the cytosolic pool of Vps4 is reduced and that fusion of 'older' MVBs with the vacuole then leaves only empty, 'younger' MVBs, with incomplete buds (*Figure 5d*). The timing for this sequence of events is consistent with our observation that Mup1-eGFP can reach the vacuole from the plasma membrane less than 15 min after methionine addition (see also *Figure 1—figure supplement 2* and *Video 8*), and with earlier work (*Arlt et al., 2015*) showing that alpha-factor also trafficked within 15 min from the plasma membrane to the vacuole. At the 120 min point, we found few endosomes with large budding structures and with very few ILVs (*Figure 5—figure supplement 1d*) or flat cisternal membrane stacks, reminiscent of class E compartments (*Figure 5d*). Thus, abrupt loss of free Vps4 stalled ILV budding and scission even though the dynamic recruitment of ESCRT-III persisted. Similar results to the 120 min time point (*Figure 5—figure supplement 1d*) were observed under long-term conditions in cells expressing mutants of Snf7 and Vps2 unable to interact with Vps4 including the appearance of a small number of endosomes with large budding structures mostly devoid of ILVs (*Adell et al., 2014*). The outcome of these long-term steady-state perturbations could not, however, define the critical step(s) for which Vps4 was required. The most straightforward molecular interpretation of our new results is that several active Vps4 hexamers participate in a single budding and fission event and that an incomplete bud will stall if no more Vps4 can add.

## Discussion

Current models for ILV formation assume gradual accumulation of ESCRT-III proteins, followed by a burst of Vps4 at the end of the process (*Schöneberg et al., 2017*). The models invoke the properties of ESCRT-III polymerization linked to a membrane surface as the driving force for membrane deformation; free energy input from ATP hydrolysis by Vps4 then re-sets the system by recycling the ESCRT-III components. Our live-cell imaging data are not consistent with this proposed sequence of events, nor are they easily reconciled with the assumption that ATP hydrolysis is needed only to complete the assembly-disassembly cycle. We summarize first our principal findings and then offer a possible alternative model.

First, appearance and disappearance of Snf7 and Vps4 at peripheral endosomes correlate, as do appearance and disappearance of Vps24 and Vps4. Therefore Snf7 and Vps24 must also correlate. Moreover, if Vps24 and Vps2 are recruited jointly, as earlier evidence suggests (*Teis et al., 2008*; *Babst et al., 2002*), then all these three ESCRT-III components will co-assemble, presumably into filamentous structures related to the diverse forms seen in vitro. Although Vps24 and Vps2 are sometimes called 'capping proteins', because they are required for the recruitment of Vps4 and thus restrict the growth of Snf7 filaments in vivo (*Teis et al., 2008*; *Babst et al., 2002*) and in vitro (*Mierzwa et al., 2017*) our data from LLSM show coordinated, rather than sequential recruitment.

Second, during the lifetime of ESCRT-III assemblies on a peripheral endosome, which ranges from 3 s to a substantial fraction of a minute, the average numbers of Snf7 and Vps24 subunits present during the event, subject to uncertainties about the incorporation ratio of tagged and untagged Snf7, are about 60 to 150 Snf7 and 14 to 30 Vps24, respectively (representing 20th to 80th percent of the distributions, *Figure 4—figure supplement 1c*). Initial recruitment is abrupt, rather than gradually cumulative. Termination is likewise abrupt. Both components fluctuate substantially, with no evident pattern, during the lifetime of the presumptive nascent bud. The fluctuations, like the initial accumulation, are abrupt; the magnitudes of these fluctuations (*Supplementary file 2*) can be nearly as great as the overall averages.

Third, in the perivacuolar endosomes, spatially unresolved in the optical microscope, large fluctuations in Snf7, Vps24 and Vps4 are of the same magnitude as a complete peripheral event (i.e. about 100–150 Snf7, 30–70 Vps24 and 20–50 Vps4 (representing 20th to 80th percent of the distributions,

*Figure 4—figure supplement 1b*), suggesting that the mechanism of ILV formation in the perivacuolar region is the same as in the periphery.

Fourth, Vps4, although it appears together with the other components, is not required for recruitment of Snf7 and Vps24. ESCRT-III recruitment did not cease under conditions of acute Vps4 depletion, and ESCRT-III recruitment kinetics were essentially the same in cells expressing ATPase impaired Vps4 as in cells expressing wild-type Vps4.

Fifth, Vps4 is required not only for scission of nascent ILVs but probably also for invagination. Following acute depletion of Vps4, the perivacuolar endosomes that accumulate contain no ILVs and at most have only one or two invaginating buds. Were Vps4 not required for invagination, we would expect those endosomes to have many such buds. The E-compartment stacks seen in cells expressing ATPase impaired Vps4 likewise appear to be devoid of buds, suggesting that for invaginations to form, Vps4 must not only be present, but its ATPase activity must also be intact. This conclusion also follows from our observation, that accumulation of more than 2–3 hexamers requires an ATPase competent Vps4. That is, an active ATPase (and, by inference, ATP hydrolysis) is necessary for a nascent ILV to proceed from an early to a later stage. (See also item (7), below).

Sixth, Vps4 is largely monomeric in the cytosol, but rapidly forms hexamers when recruited along with Snf7 and Vps24 to endosomes. Fluctuations in Vps4 content during the lifetime of a recruitment event were generally in multiples of six.

Seventh, approximately 20% of the Vps4 and ESCRT-III recruitment events are probably 'non-productive' – that is, do not result in a completed ILV. Our evidence for this conclusion is as follows. Addition of cargo led to immediate uptake by the cells mediated by their ubiquitinated receptors, presumably into small early endosomes (*Kukulski et al., 2012a*), which recruited Vps4 and Snf7. Previous work has shown, however, that only when cargo reaches larger compartments, probably generated by fusion of the smaller vesicles, do ILVs appear (*Prescianotto-Baschong and Riezman, 1998*). Indeed, early endosomes are too small to host production of ILVs, and only when they reach some critical size is it likely that ILVs can invaginate and pinch. We suggest that most of the peripheral events that accumulate three or more Vps4 hexamers represent formation of complete ILVs, as such events are absent in cells expressing the ATPase-impaired mutant and in the *pep12Δ* cells that cannot form endocytic carriers large enough to accommodate an ILV.

As this summary illustrates, we have shown that Vps4 is present throughout the process of ILV formation and that its ATPase activity is required to proceed from an early stage to a later one, as well as for vesicle scission. These findings are not compatible with current models. What sort of molecular mechanism might they suggest instead? Snf7, Vps24 and other, related ESCRT-III components all form filamentous or helical polymers in vitro (*Schöneberg et al., 2017*), but because a variety of polymorphic assemblies of these molecules have been characterized, we cannot from available data specify what structure(s) correspond to the in vivo events recorded here. The apparent co-assembly of Snf7 with Vps24 (and probably Vps2) and the estimated amounts of these ESCRT-III subunits in a typical recruitment are not, however, consistent with formation of very long, coiled structures on membranes. Moreover, rapid fluctuations in the numbers of Snf7 and Vps24 during the lifetime of an ESCRT-III mediated event suggest that two or more shorter filamentous structures may be forming, each a co-polymer of the ESCRT-III components, rather than a single continuously growing filament. Bro1, the ESCRT-0 component can nucleate ESCRT-III (*Tang et al., 2016*). In addition, Vps25, the ESCRT-II component thought to nucleate ESCRT-III through Vps20, is present in two copies (*Hierro et al., 2004*; *Teo et al., 2004*; *Teis et al., 2010*). Thus, multiple nucleations are therefore in principle possible. If the observed ESCRT-III dynamics indeed reflect growth and shrinkage (or appearance and disappearance) of more than one filament, then invagination and pinching may require some correlation of the otherwise stochastic behavior of multiple filaments, presumably brought about by the ATPase activity of several Vps4 hexamers. One picture consistent with these notions is illustrated in *Figure 6*.

The model in *Figure 6* has the following features. First, it postulates two or more ESCRT-III filaments, wrapped around the ubiquitinated cargo - ESCRT-0, I, II complexes that recruit them. Second, it postulates that the ATPase activity of Vps4 hexamers drives invagination and ultimately scission of nascent buds, through mechanochemical coupling to the ESCRT-III polymers. Vps4 hexamers interact with ESCRT-III polymers (*Shen et al., 2014*; *Ghazi-Tabatabai et al., 2008*; *Hanson et al., 2008*) through their MIT domains with the MIM elements near the C-termini of ESCRT-III subunits (*Obita et al., 2007*; *Stuchell-Brereton et al., 2007*); the interaction with Vps2

MIM elements is particularly strong (*Obita et al., 2007*; *Stuchell-Brereton et al., 2007*). When assembled as a hexamer, Vps4 can also unfold an ESCRT-III subunit by passing the ESCRT-III polypeptide chain through its central pore (*Yang et al., 2015*; *Su et al., 2017*; *Monroe et al., 2017*). Thus, there are a variety of ways in which a Vps4 hexamer, anchored on one filament, can pull on components of another. In the particular version, shown in *Figure 6*, cargo-associated ESCRT-0-I-II complexes (not displayed) nucleate rapid assembly of several ESCRT-III filaments, which grow and shrink stochastically. A Vps4 hexamer (several are generally present), docked by MIT-MIM association on one filament, exerts ATP-dependent force on another by pulling the C-terminus of an ESCRT-III subunit into the hexamer pore, crowding the membrane-anchored cargo together as the cycling process ensues during the duration of the event (cargo crowding). The crowding forces membrane invagination (membrane buckling) to create the surface area needed to accommodate the cargo molecules, which generally have larger luminal domains than cytosolic projections (*Derganc et al., 2013*; *Derganc and Čopič, 2016*; *Stachowiak et al., 2012*; *Mageswaran et al., 2015*). Thus, in this model, Vps4 drives invagination primarily by concentrating cargo, rather than by favoring transformation of an ESCRT-III filament from a planar coil to dome-like one, as in several other proposed models. The capacity to adopt a dome-like curvature (*Lata et al., 2008*) will, however, allow ESCRT-III filaments to accommodate to an invaginating bud, probably increasing the capacity for Vps4 hexamers, as observed for productive events. The forces exerted by the interactions between ESCRT-III and Vps4 also generate the neck of the nascent bud (ILV budding), leading to neck tightening, membrane constriction and ultimately to vesicle fission and final release of the ESCRT machinery. Many variations of this picture are consistent with our current data.

## Materials and methods

### Antibodies and reagents

The following antibodies were used for western blot analysis with the stated dilutions: rabbit polyclonal antiserum (1:200) specific for Vps2 (SAT 455 alpha) purified as described (*Adell et al., 2014*); polyclonal rabbit antisera (1:5000) specific for Vps4, Snf7 and Vps24 were a gift from the Emr Lab (*Teis et al., 2008*); mouse monoclonal antibody (1:10000) specific for PGK (phosphoglycerate kinase 1, (Invitrogen, Rockford, Illinois, USA, 459250); mouse monoclonal antibody (1:1000) specific for GFP (IgG1K, Roche Diagnostics, Germany, 11814460001, RRID: AB_390913); goat anti-mouse IgG–Peroxidase (Sigma, St. Louis, Missouri, USA, A4416, 1:5000, RRID: AB_258167); goat anti-Rabbit IgG–Peroxidase (Sigma, St. Louis, Missouri, USA, A0545, 1:5000). FM4-64 for life cell microscopy was purchased from Invitrogen (United Kingdom, T-3166); Rapamycin from LC Laboratories (Woburn, Massachusetts, USA, R-5000) and Concanavalin A from Canavalia ensiformis from Sigma (St. Louis, Missouri, USA, L7647). For Canavanine sensitivity assays we used L-Canavanine from Sigma (St. Louis, Missouri, USA, C1625). TetraSpeck Microspheres, 0.05 µm (Thermo Scientific, Eugene Oregon, USA C47281) were used for correlative fluorescence and electron microscopy. Goat anti-GFP (Rockland, Limerick, Pennsylvania, USA, 600-101-215, 1:500, RRID: AB_218182), visualized by rabbit anti-goat Fab' NANOGOLD (Nanoprobes, Yaphank, New York, USA, 2004, 1:150) plus silver enhancement with HQ-Silver (Nanoprobes, Yaphank, New York, USA, 2012) was used for immunogold labeling. Additional Information on Antibodies and Reagents is provided in *Supplementary file 3*.

### Strains, Plasmids and DNA manipulation

*Saccharomyces cerevisiae* strains were grown to mid-log phase in YNB medium without methionine (unless otherwise indicated). Strains, plasmids and oligonucleotides are described in *Supplementary file 3*. The mNeonGreen Advanced vector was purchased from Allele-Biotechnology (purchase license L. A. Huber, Division of Cell Biology, Medical University Innsbruck). All plasmids were generated by standard cloning techniques. The shuttle vectors used have been described (*Adell et al., 2014*).

### Preparation of yeast whole cell protein extracts

Log phase yeast cells were pelleted, resuspended in ice-cold water with 10% trichloroacetic acid (TCA), incubated on ice for at least 30 min and washed twice with acetone. The precipitate was

resolubilized in boiling buffer (50 mM Tris-HCl, pH 7, 5; 1 mM EDTA, 1% SDS), solubilized with glass beads and boiled at 95°C. Urea sample buffer (150 mM Tris-HCl, pH 6, 8, 6 M Urea, 6% SDS, bromophenol blue, 10% β-mercaptoethanol) was added and the cleared cell lysate analyzed by SDS-PAGE.

## Subcellular fractionation

Subcellular fractionation of proteins into membrane-associated pellet and soluble cytoplasmic fractions was performed from mid-log cells as described (*Babst et al., 1997*). In vivo DSP crosslinking (*Figure 4—figure supplement 10*) and cell lysis of yeast cells expressing Vps4-eGFP was performed as described (*Copic et al., 2007*).

## Canavanine sensitivity assay

Yeast cells were grown over night to log phase $OD_{600}$ ~0.6 and serial dilutions were spotted on agar plates with YPD complete medium, 1x YNB selective medium and 1x YNB + 1µg/ml L-Canavanine as described (*Teis et al., 2010*).

## Inducible in vivo heterodimerization by rapamycin ('anchor-away')

Pma1 was chosen as an anchor because it is about 200x more abundant when compared to Vps4 (*Ghaemmaghami et al., 2003*) and has been used as a well characterized anchor many times before (e.g. *Haruki et al., 2008*). Moreover Pma1 is a stable cell surface protein with a half-life of ~11 hr (*Benito et al., 1991*), (*Shih et al., 2000*). Heterodimerization of FKBP12 with FRB fused to Vps4 was achieved in yeast cells grown to mid-log phase and then treated with 1 µg/ml rapamycin (rapamycin stock = 50 mg/ml in 100% EtOH) . Vps4-(eGFP)-FRB was efficiently anchored way in less than one minute and remained inactivated for the duration of the experiments (ranging from 30 s to 2 hr). When Vps4-GFP-FRB was co-expressed with Vps4-mCherry in the same cell, Vps4-mCherry remained in the cytoplasm and on endosomes. It was not co-recruited by Vps4-eGFP-FRB to Pma1-FKBP1 at the plasma membrane. This result suggested that once Vps4-eGFP-FRB was anchored to Pma1-FKB12 it could no longer form functional complexes (*Figure 5—figure supplement 1*). Under these conditions Vps4-eGFP-FRB was only visible at the PM, did not accumulate on endosomes and was not transported into the vacuole via the MVB. When Vps4-eGFP-FRB was co-expressed with untagged Vps4, MVB sorting remained active (*Figure 5a*) after anchoring away Vps4-eGFP-FRB. Vps4-GFP-FRB remained anchored to Pma1-FKBP12, it remained at the PM and was not detected on endosomes and it was not transported to the vacuole for the duration of the experiment – unlike Mup1-mCherry that was efficiently delivered to the vacuole in the same cells.

## Epifluorescence microscopy

Live cell epifluorescence microscopy was carried out using a Zeiss Axio Imager M1 equipped with a SPOT Xplorer CCD camera, standard fluorescent filters and AxioVision software. Exposure time was 500 msec for all fluorescent channels (eGFP, mCherry, mRuby2, FM4-64) and 50 msec for phase contrast images. Membrane labeling by FM4-64 dye (Invitrogen) was done in midlog ($OD_{600}$ = 0.6) cells (*Teis et al., 2008*; *Vida and Emr, 1995*). Cells were incubated with FM4-64 dye for 5 min at 26°C, washed twice with YNB, re-suspended in selection medium, incubated for one hour at 26°C and then imaged (*Vida and Emr, 1995*; *Teis et al., 2008*). The brightness and contrast of the images in the figure were adjusted using Photoshop CS5 (version 12.0.4 × 64; Adobe).

## Fluorescence recovery after photobleaching

Live cell confocal microscopy was performed using a TSC SP5 confocal laser-scanning microscope (Leica) and a 63x Leica Objective (HC-PL-APO-CORR-CS2, NA = 1,20), with cells grown to midlog ($OD_{600}$ = 0.4–0.6) labeled with FM4-64 5 min before imaging. Cells were mounted on concavalin A coated cover slides. FRAP was carried in perivacuolar regions of interest containing Vps4-eGFP and FM4-64; bleaching was achieved by 10 ms exposure of an Argon laser beam (digital power 40%, intensity 40%) emitting at 488 nm (point-bleaching setting, Leica-FRAP Wizard). Vps4-eGFP was imaged with 488 nm (10% intensity, HyD 100% gain), FM4-64 was imaged with 561 nm (3% intensity, HyD detector 100% gain). The time-series were acquired with an imaging speed of 1400 Hz. For analysis, fluorescence intensity was corrected for background and bleaching, and normalized to the

fluorescence intensity from the first frame. The rate of recovery after photobleaching and the fitted time constants were determined using a single-exponential fitting custom-made MATLAB script.

## Correlated light and electron microscopy

Correlative light and electron microscopy was performed as described with minor modifications (*Kukulski et al., 2011*; *Suresh et al., 2015*). Briefly, Vps4-eGFP expressing cells were grown in SD-Met media and Vps4-mNeonGreen expressing cells in SC-Trp at 30°C. Yeast cells were harvested by filtration and then subjected to high-pressure rapid freezing. Samples were freeze-substituted and then embedded in Lowicryl HM20. 200–300 nm thick sections were collected on copper grids with a continuous carbon film. TetraSpeck beads of 50 nm diameter used as markers for the correlation procedure were deposited on the sample surface. The grids were imaged on an Olympus IX81 inverted microscope equipped with 100×, N/A 1.45 objective and an Orca-ER camera (Hamamatsu). Three channels were acquired for each field of view (blue with excitation filter 377/50 nm and emission filter 473/30 nm; green with excitation filter 470/22 nm and emission filter 520/35 nm; red with excitation filter 556/20 nm and emission filter 624/40 nm). Before electron tomography 15 nm fiducial gold conjugates were applied to the section surface as fiducial markers for tomogram alignment, and samples were contrasted with Reynolds lead citrate. Electron tomography was performed using Tecnai F30 (FEI) transmission electron microscope at the EMBL Electron Microscopy Core Facility, operated at 300 KV, equipped with a FEI Eagle 4K CCD camera and dual tilt holder. Data were collected using the SerialEM software (*Mastronarde, 2005*). Two tomograms were recorded for each area of interest: a dual-axis high-magnification tomogram with pixel size 1.202 nm, tilt range −60° to +60°, and 1° tilt angle increment and a low magnification tomogram with a pixel size of 2.59 nm, tilt range −60° to +60°, and a 2° tilt angle increment.

## Sample preparation for tomography and immuno-gold labeling

High-pressure freezing and freeze-substitution was performed as described (*Adell et al., 2014*; *Schmiedinger et al., 2013*). For tomography, samples were subsequently embedded in epoxy resin and 100–300 nm sections analyzed after post-staining. Electron tomography from 300 nm sections (coated with 10 nm fiducial gold conjugates) was performed on a Tecnai T20-G2 (FEI) operated at 200kV using dual-tilt series. Images were recorded at binning 2 with a 4 × 4 k Eagle digital camera (FEI) from 55° to −55° with 1° increments using Inspect3D automated tomography software (FEI). For immunoelectron microscopy, freeze-substituted samples were rehydrated (*van Donselaar et al., 2007*), and processed for indirect immunogold labeling (*Schmiedinger et al., 2013*) of 100 nm-thick, thawed cryosections.

## Tomographic reconstructions and analysis

Tomogram reconstruction, 3D modeling and analysis were performed using the IMOD software package (*Kremer et al., 1996*). The analysis for correlating light and electron microscopy was performed as described using a Matlab-7.4 script (*Kukulski et al., 2012b*).

## Real time 3D LLSM

Yeast cells expressing fluorescently tagged ESCRT-III, Vps4 and cargo proteins were grown to log phase in selective SC media. Cells were concentrated by centrifugation and $2 \times 10^7$ cells re-suspended in 10 μl selective SC media and spotted on top of concanavalin A-coated 5 mm round glass coverslips. After 5 min, coverslips were placed in the sample bath of our LLSM (*Aguet et al., 2016*) containing selective SC media. The samples were imaged as a time series in 3D using a dithered multi-Bessel lattice light-sheet by stepping the sample stage at 500 nm intervals in the s-axis equivalent to ~261 nm translation in the z-axis (*Figure 3a*); thus, each 3D image took 850 ms to acquire (including a 100 ms pause between imaging volumes) for a total of 60 time points. Each 3D stack corresponded to a pre-deskewed volume of ~50 μm x 50 μm x 15 μm (512 × 512×30 pixels). The inner and outer NAs of excitation were 0.505 and 0.6, respectively. Yeast cells only expressing Vps4-eGFP, Vps4$^{E233Q}$-eGFP, or Vps4-eGFP in the *pep12Δ* background, were excited with a 488 nm laser (~120 mW operating power with an illumination of ~300 μW at the back aperture) to acquire 28–56 imaging planes, each exposed for ~14.8–21 ms and recorded using an Andor iXon 897 EMCCD camera. Same conditions were used to image cells expressing Snf7-eGFP in the *pep12Δ* background.

Cells expressing combinations of Snf7-eGFP and Vps24-eGFP with either Vps4-mCherry or Vps4$^{E233Q}$-mCherry were sequentially excited with a 488 nm laser (~240 mW power and ~600 µW at back aperture) and a 561 nm laser (~500 mW power and ~400 µW at back aperture) for each optical plane. Images for each channel were recorded with two Andor iXon 897 EMCCD cameras using exposures of ~11.8 ms per channel. These imaging conditions provided with the sensitivity sufficient to detect the fluorescence signal with a signal-to-noise ratio of 3–5 from 3 eGFP molecules located within a diffraction-limited spot. Under these imaging conditions used for cells expressing the Vps4$^{E233Q}$ mutants, the dynamic range of the EMCCD became saturated for the fluorescence signal elicited by ESCRT-III subunits or Vps4 associated with the E compartment (*Figure 3*, *Figure 3—figure supplement 1*).

## Analysis of ESCRT-III and Vps4 recruitment dynamics

### 3D image preprocessing

Approximately 300–1000 yeast cells were analyzed per experimental condition. The 3D stacks were first cropped so as to only include data illuminated by the non-diffracting region of the light-sheet, then flat-field corrected to normalize for illumination, and finally deskewed using a geometric image transform function as described (*Aguet et al., 2016*). The diffraction-limited spots were then detected and tracked in three dimensions using the same automated algorithms developed to follow the formation of clathrin-coated structures in the entire volume of a cell (*Aguet et al., 2016*). The MATLAB implementations of LLSM volume deskewing, point-source detection and 3D tracking software are available for download at https://github.com/francois-a/llsmtools (*Aguet et al., 2016*). Valid tracks used for further analysis encompassed those that did not merge or split during the duration of the object, whose lifetimes were fully included in the time series, whose fluorescence intensity where above the local background and sensitivity threshold and whose positions were farther than 1.5 µm from any edge of the imaged volume. Persistent tracks also used for analysis corresponded to those with lifetimes longer than the time series.

### Single molecule eGFP fluorescence calibration

The LLSM was calibrated for three-molecule eGFP detection using bacterially produced eGFP adsorbed to a glass coverslip and imaged with the LLSM (*Figure 4—figure supplement 10e*). The microscope was adjusted to detect the fluorescence of single eGFP molecules, thus allowing us to generate a calibration curve obtained with different exposures and with the same laser power conditions used to image the cells. A custom-made MATLAB (MathWorks) script automatically detected the 3D-fitted asymmetric Gaussian fluorescence

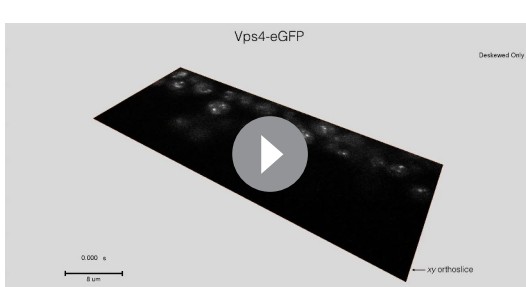

**Video 1.** Dynamics of Vps4-eGFP recruitment to endosomes of yeast cells visualized using LLSM. Related to *Figure 3f*. The movie starts by showing the acquisition of sequential raw, non-deskewed imaging planes acquired every 20 ms using 3D LLSM from live cells expressing Vps4-eGFP. Then it shows the appearance of the same imaging planes after 3D deconvolution and rendering. It ends with a 3D time series including orthogonal deconvolved side views lasting 51 s (also shown in *Video 2*).

DOI: https://doi.org/10.7554/eLife.31652.006

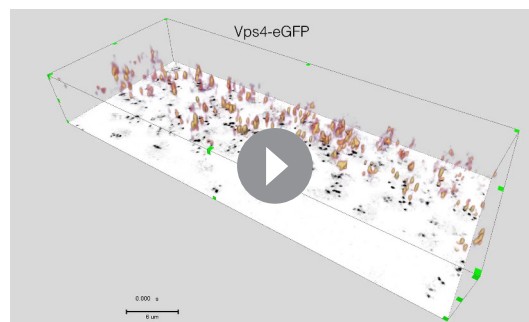

**Video 2.** Dynamics of Vps4-eGFP recruitment to endosomes of yeast cells visualized using LLSM. Related to *Figure 3*. Cells expressing Vps4-eGFP were imaged for 51 s. The movie shows a deconvolved and rendered 3D view and orthogonal deconvolved side views. The fluorescent objects were traced using the automated 3D detection and tracking software and then color coded as rainbow from blue to red following strength of the signal.

DOI: https://doi.org/10.7554/eLife.31652.007

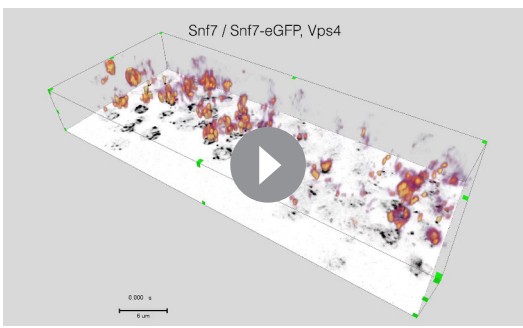

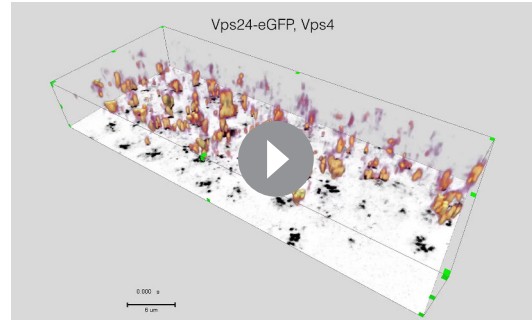

**Video 3.** Dynamics of Snf7-eGFP recruitment to endosomes of yeast cells visualized using LLSM. Related to *Figure 3*. Cells expressing a mixture of Snf7 and Snf7-eGFP together with Vps4-mCherry were imaged for 51 s. For simplicity, the movie only shows the fluorescence for Snf7-eGFP, although all the fluorescence signals of Snf7-eGFP and Vps4-mCherry colocalized (*Figure 3*). The movie shows a deconvolved and rendered 3D view and orthogonal deconvolved side views. The fluorescent objects were traced using the automated 3D detection and tracking software and then color coded as rainbow from blue to red following strength of the signal.
DOI: https://doi.org/10.7554/eLife.31652.008

**Video 4.** Dynamics of Vps24-eGFP recruitment to endosomes of yeast cells visualized using LLSM. Related to *Figure 3*. Cells expressing Vps24-eGFP together with Vps4-mCherry were imaged for 51 s. For simplicity, the movie only shows the fluorescence for Vps24-eGFP, although all the fluorescence signals of Vps24-eGFP and Vps4-mCherry colocalized (*Figure 3*). The movie shows a deconvolved and rendered 3D view and orthogonal deconvolved side views. The fluorescent objects were traced using the automated 3D detection and tracking software and then color coded as rainbow from blue to red following strength of the signal.
DOI: https://doi.org/10.7554/eLife.31652.009

intensity associated with a diffraction-limited spot; a T-test was used to identify valid spots whose 3D-fitted fluorescence signal was statistically higher than their local background before bleaching. The fluorescence intensity distribution was then fitted to a mixture-model Gaussian function (*Aguet et al., 2013*), where the first Gaussian population corresponded to signal elicited by a single-eGFP molecule. The eGFP fluorescence signals detected in the yeast cells were then

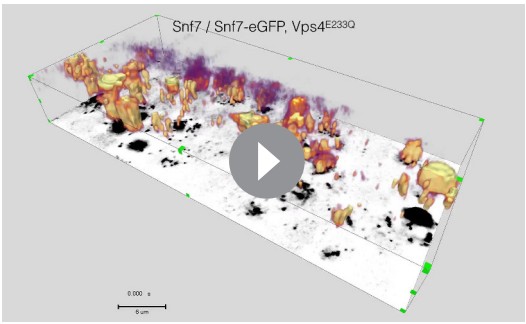

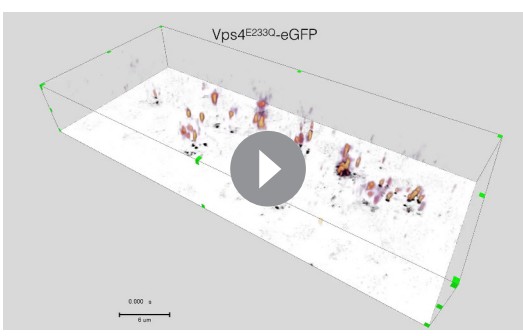

**Video 5.** Dynamics of Vps4$^{E233Q}$-eGFP recruitment to endosomes of yeast cells visualized using LLSM. Related to *Figure 3*. The 51 s 3D time series acquired using LLSM of yeast cells expressing Vps4 $^{E233Q}$-eGFP. The movie shows a deconvolved and rendered 3D view and orthogonal deconvolved side views. The fluorescent objects were traced using the automated 3D detection and tracking software and then color coded as rainbow from blue to red following strength of the signal.
DOI: https://doi.org/10.7554/eLife.31652.010

**Video 6.** Effect of the Vps4$^{E233Q}$ mutant on the dynamics of Snf7-eGFP recruitment to endosomes of yeast cells visualized using LLSM. Related to *Figure 3*. Cells expressing a mixture of Snf7 and Snf7-eGFP together with Vps4$^{E233Q}$-mCherry were imaged for 51 s. Although all the fluorescence signals of Snf7-eGFP and Vps4$^{E233Q}$-mCherry colocalized (*Figure 3*), for simplicity, the movie only shows the fluorescence for Snf7-eGFP. The movie shows a deconvolved and rendered 3D view and orthogonal deconvolved side views. The fluorescent objects were traced using the automated 3D detection and tracking software and then color coded as rainbow from blue to red following strength of the signal.
DOI: https://doi.org/10.7554/eLife.31652.011

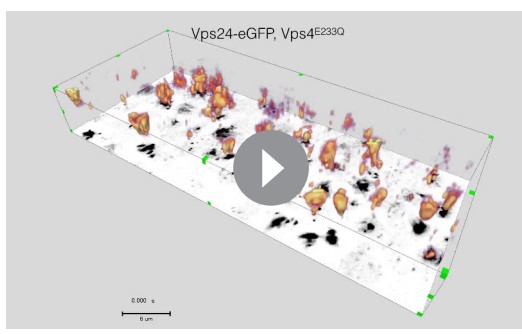

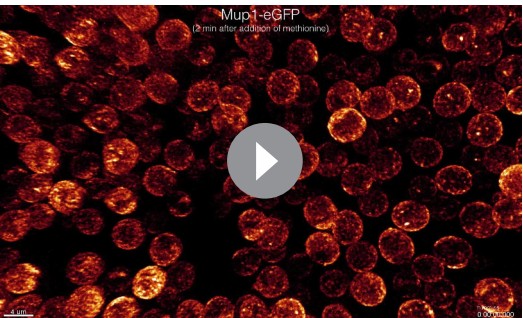

**Video 7.** Effect of the Vps4[E233Q] mutant on the dynamics of Vps24-eGFP recruitment to endosomes of yeast cells visualized using LLSM. Related to *Figure 3*. Cells expressing Vps24-eGFP together with Vps4[E233Q]-mCherry were imaged for 51 s. Although all the fluorescence signals of Vps24-eGFP and Vps4[E233Q]-mCherry colocalized (*Figure 3*), for simplicity, the movie only shows the fluorescence for Vps24-eGFP. The movie shows a deconvolved and rendered 3D view and orthogonal deconvolved side views. The fluorescent objects were traced using the automated 3D detection and tracking software and then color coded as rainbow from blue to red following strength of the signal.
DOI: https://doi.org/10.7554/eLife.31652.012

**Video 8.** Mup1-eGFP endocytic transport visualized using LLSM. Related to *Figure 3*. Live cells expressing the methanione transporter Mup1-eGFP were imaged using LLSM for 20 min starting 2 min after the addition of methionine. The movie shows 3D orthogonal deconvolved and rendered views; time points were acquired every 5 s for a total of 20 min. Mup1-eGPF accumulates in the perivacuolar region in 5–10 min.
DOI: https://doi.org/10.7554/eLife.31652.026

converted to number of eGFP molecules and the traces include the propagated error (square root of the squared sums of uncertainties from the 3D fitted fluorescence signal and from the single-eGFP calibration curve). The spinning disc confocal microscope was calibrated for single-molecule eGFP as described above but using a 2D Gaussian fluorescence intensity-fitting algorithm (*Cocucci et al., 2012*) (*Aguet et al., 2013*).

## Fluctuation dynamics, lifetimes, number of molecule and visualization

Imaging of diffraction limited fluorescent beads illuminated so as to generate the signal equivalent to ~50 eGFPs showed constant fluorescence signal with a standard deviation of ~3; this indicates that the large signal fluctuations determined in the yeast cells were not due to the instrument but instead were biological fluctuations due to change in the number of fluorescence molecules recruited to a single spot. The same conclusion was reached by demonstrating absence of fluorescence intensity fluctuations in time series of Mup1-eGFP cargo internalized into the diffraction-limited peripheral endosomes in yeast cells imaged with the same conditions of illumination and acquisition used to trace ESCRT-III and Vps4 dynamics. Correlation dynamics (*Figure 4a*), lifetimes (*Figure 4b*) and maximum number of molecules accumulated within a trace (*Figure 4c,d* and *Figure 4—figure supplement 1a*) were obtained using custom-made MATLAB routines. The plots in *Figure 4a* show the cross-correlation of fluorescence data and the corresponding derivative obtained in the eGFP and mCherry channels for traces lasting 11 s or more. The plots in *Figure 4b–d* and *Figure 4—figure supplement 1* show probability density distributions and cumulative distributions; the last bin in *Figure 4c* and *Figure 4—figure supplement 1* show the 95th – 100th percentile of values. The local temporal fluctuations in the number of molecules associated with perivacuolar objects were determined using the findpeaks function included in a custom-made MATLAB script. 3D volume rendered images (*Figure 3f*, *Figure 3—figure supplement 1*) and movies were generated using Amira 6.1–6.3 (FEI, Thermo Fisher Scientific). When indicated, movies were deconvolved using the Lucy–Richardson algorithm (deconvlucy function in MATLAB) by providing measured background and an experimentally measured PSF for 15 iterations.

## Statistical analysis

Statistical Analyses were performed using two-sample Kolmogorov-Smirnov test (*Figure 4b–d*, *Figure 3—figure supplement 1*), and Student's T-test (*Figure 4—figure supplement 10b,d*), two-sample permutation test for means or medians (*Figure 4b,c*, *Figure 5—figure supplement 1c*).

## Acknowledgements

We thank H Bergler for yeast strains, M Offterdinger for help with laser confocal microscopy, JR Houser from maintaining the spinning disc microscope, E Cocucci for advice setting up the single molecule calibration by spinning disc microscopy, and K Gutleben and B Witting for EM-sample preparation. SU thanks H Elliott, D Richmond and D Hoffman for discussions and the MATLAB code repository received from the Computational Image Analysis Workshop supported by NIH grant GM103792. We specially thank SC Harrison for discussions and editorial help. TK acknowledges support from the Janelia Visitor Program and Eric Betzig, Eric Marino, Tsung-Li Liu and Wesley R Legant for help and advice in constructing and installing the lattice light-sheet microscope. Construction of the lattice light-sheet microscope was supported by grants from Biogen and Ionis Pharmaceuticals to TK The research was supported by a NIH grant R01 GM075252 to TK, the Austrian Science Fund (FWF-Y444-B12, P30263) to DT, MCBO (W1101-B18) to DT, Deutsche Forschungsgemeinschaft SFB 1129 Z2 to JAGB and an Austrian Marshall Plan Scholarship to MP. SU is a Fellow at the Image and Data Analysis Core at Harvard Medical School. None of the authors have a financial interest related to this work.

## Additional information

### Funding

| Funder | Grant reference number | Author |
| --- | --- | --- |
| Deutsche Forschungsgemeinschaft | SFB 1129 Z2 | John AG Briggs |
| National Institutes of Health | GM075252 | Tomas Kirchhausen |
| Biogen Idec | | Tomas Kirchhausen |
| Ionis Pharmaceuticals | | Tomas Kirchhausen |
| Austrian Science Fund | Y444-B12 | David Teis |
| Austrian Science Fund | P30263 | David Teis |
| Austrian Science Fund | W1101-B18 | David Teis |
| Austrian Marshall Plan Foundation | Austrian Marshall Plan Scholarship | Mehrshad Pakdel |

The funders had no role in study design, data collection and interpretation, or the decision to submit the work for publication.

### Author contributions

Manuel Alonso Y Adell, Simona M Migliano, Formal analysis, Validation, Investigation, Visualization, Methodology, Writing—original draft; Srigokul Upadhyayula, Conceptualization, Software, Formal analysis, Validation, Investigation, Visualization, Methodology, Writing—original draft; Yury S Bykov, Formal analysis, Investigation, Visualization, Methodology, Writing—original draft; Simon Sprenger, Formal analysis, Investigation, Methodology; Mehrshad Pakdel, Gloria Jih, Reza Behrouzi, Markus Babst, Investigation, Methodology; Georg F Vogel, Investigation, Visualization, Methodology; Wesley Skillern, Formal analysis,; Oliver Schmidt, Resources, Investigation, Methodology; Michael W Hess, Supervision, Validation, Investigation, Visualization, Methodology, Writing—original draft; John AG Briggs, Formal analysis, Supervision, Funding acquisition, Validation, Investigation, Writing—original draft; Tomas Kirchhausen, Conceptualization, Formal Analysis, Supervision, Funding acquisition, Validation, Investigation, Visualization, Methodology, Writing—original draft, Project administration, Writing—review and editing; David Teis, Conceptualization, Formal analysis,

Supervision, Funding acquisition, Validation, Investigation, Visualization, Writing—original draft, Project administration, Writing—review and editing

## Author ORCIDs

Simona M Migliano https://orcid.org/0000-0002-1888-5332
Srigokul Upadhyayula https://orcid.org/0000-0002-6911-0270
Yury S Bykov https://orcid.org/0000-0003-2959-4108
Reza Behrouzi http://orcid.org/0000-0003-3064-9743
Tomas Kirchhausen https://orcid.org/0000-0003-0559-893X
David Teis http://orcid.org/0000-0002-8181-0253

## Decision letter and Author response

Decision letter https://doi.org/10.7554/eLife.31652.036
Author response https://doi.org/10.7554/eLife.31652.037

# Additional files

## Supplementary files

• Supplementary file 1. Statistics of the Vps4-eGFP and Vps4-mNeonGreen CLEM dataset
DOI: https://doi.org/10.7554/eLife.31652.030

• Supplementary file 2. Summary of the quantitative data for Snf7-eGFP, Vps24-eGFP and Vps4-eGFP in WT cells and in the respective mutants.
DOI: https://doi.org/10.7554/eLife.31652.031

• Supplementary file 3. This table contains information on yeast strains, plasmids, DNA primer, Antibodies, chemical reagents and software used in this work.
DOI: https://doi.org/10.7554/eLife.31652.032

• Transparent reporting form
DOI: https://doi.org/10.7554/eLife.31652.033

## Major datasets

The following dataset was generated:

| Author(s) | Year | Dataset title | Dataset URL | Database, license, and accessibility information |
|---|---|---|---|---|
| Adell M, Migliano S, Upadhyayula S, Bykov Y, Sprenger S, Pakdel M, Vogel G, Jih G, Skillern W, Behrouzi R, Babst M, Hess M, Briggs J, Kirchhausen T, Teis D | 2017 | Data from: Recruitment dynamics of ESCRT-III and Vps4 to endosomes and implications for reverse membrane budding | http://dx.doi.org/10.5061/dryad.gn250 | Available at Dryad Digital Repository under a CC0 Public Domain Dedication |

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
