## [Decision Letter]

Congratulations, we are pleased to inform you that your article, "Recruitment dynamics of ESCRT-III and Vps4 to endosomes and implications for reverse membrane budding", has been accepted for publication in *eLife*. Your article has been reviewed by three peer reviewers, one of whom is a member of our Board of Reviewing Editors, and the evaluation has been overseen by Vivek Maholtra as the Senior Editor.

This work is the first rigorous examination of ESCRT-III and Vps4 dynamics in cells as these molecules generate endosome intralumenal vesicles. Contrary to prevailing models, which propose a sequence of ESCRT-III filament polymerization that drives ILV budding and concludes with Vps4 recruitment and ILV fission elicited by Vps4 ATPase activity, this manuscript shows that Vps4 and ESCRT-III associate with the endosome continuously, that Vps4 ATPase activity correlates with ESCRT-III dynamics, and that Vps4 is required for ILV formation, not just fission. The authors propose a new model that accounts for the observed dynamics that will be of great interest to the field.

The reviews have been included at the end of this letter. As you will see, the reviewers made a few suggestions for experiments that might strengthen and/or extend your findings, however, addressing these points is optional. Please let us know if you wish to consider any of these new experiments as it would then be necessary to consult the reviewing editor for approval of new additions.

Reviewer #1:

A study of ESCRT protein dynamics on endosomes of live cells is presented. Studies of purified and in vivo overexpressed ESCRT III proteins show that Snf7 and other ESCRT-III subunits are capable of forming spiral filaments that deform membrane; these filaments are proposed to entrap integral membrane proteins and to possess intrinsic membrane bending activity. Prevailing models of ILV biogenesis also propose that the ATPase cycle of Vps4 elicits ESCRT-III filament disassembly and release of ESCRT-III proteins from the membrane, concomitant with vesicle fission. In this study, cutting edge optical and electron microcopy tools were implemented to test several assumptions of the prevailing models by monitoring the association of fluorescently tagged ESCRT-III and Vps4 molecules with yeast endosomes, over time, in live cells. The authors report that ESCRT-III and Vps4 dynamics are inconsistent with current models that posit sequential accumulation of ESCRT-III molecules (i.e., filament growth) that drives ILV membrane budding, concluding with the action of Vps4 to drive ILV fission. Based on their results, a conceptually new mechanistic model is proposed in which continuous association/dissociation of ESCRT-III and Vps4 molecules with the maturing endosome membrane elicits the formation of a dynamic filamentous network that concentrates integral membrane proteins to drive membrane bending via molecular crowding, culminating in vesicle fission. Overall, this is a nicely executed, interesting study that, despite a few technical caveats, is certain to stimulate an important debate within the field.

1) Ambiguity regarding the presumed mixed tagged-untagged Snf7 filaments on the endosome is concerning because the degree to which the tag changes Snf7 behavior (i.e., kinetics of filament assembly) is not established. Affected protein functions, if any, might significantly skew the interpretation of the data by restricting analyses to events that do not reflect the physiological situation. In addition, if the assumption that tagged and untagged Snf7 are incorporated with equal probability into an ESCRT-III filament is not correct, then the estimate of Snf7 abundance on the endosome is compromised. I wonder if simple Snf7 fractionation experiments to determine the proportion of tagged Snf7 in membrane-associated cell membrane fraction might lend some clarity to this question and confidence to the authors' assumption.

2) The small proportion of endosomes for which Vps4-mCherry could be detected and analyzed (~30%) begs the question, are the analyzed events representative? Similar to the issue raised in point 1, if the assumption that tagged Vps4 function equivalently to untagged Vps4 (the included tests of tagged Vps4 function are not sufficiently quantitative to determine this), then it is difficult to gauge the significance of restricting the analysis to the subset of endosomes that contain enough Vps4-mCherry to analyze. Have the authors tested if other tags (e.g., halotag) increase Vps4 detection sufficiently to analyze a greater proportion endosomes?

Reviewer #2:

Teis, Kirchhausen, et al. use a number of new and powerful approaches to provide a detailed view of ESCRT machinery dynamics and function during yeast MVB biogenesis. This work – enabled largely by lattice sheet microscopy – is the first really the first serious look at ESCRT-III and Vps4 dynamics as they generate intralumenal vesicles, and will be of great interest to the field. The data is dense but clearly presented, with balanced interpretations that should help to stimulate new thinking in the field. I recommend publication.

1) The authors might clarify the distinction they are making between short and long ESCRT-III polymers; a more or less linear assembly of subunits in ESCRT polymers should allow for either with the observed number of subunits.

2) The VPS4EQ data and interpretation is somewhat difficult to follow; the authors should clarify their description of this data and the comparison to the knock-sideways results.

Reviewer #3:

This represents a carefully conducted study that uses quantitative microscopy to address existing models of ESCRT-III assembly, Vps4 recruitment, and ILV formation during MVB sorting. The data supports a conclusion that we need to reconsider existing models pertaining to how these proteins function together and this is valuable as there have been pieces of data in the literature for some time that suggested as much. Data comes from high end instrumentation applied in a manner previously unrealized, is presented in a logical and level manner. While it would appear to be inconsistent with aspects of the existing model it is not clear that it has blown that model out of the water. Nor does it test the new model. Yet there seems little doubt this will drive additional experiments and conversations within the field. I am supportive of publishing this work because I believe the present model is limiting our progress.

One thought that comes to mind is that the model puts multiple Vps4 oligomers at the site of ILV formation. A factor that has been implicated in stabilizing interactions between Vps4 oligomers is Vta1. This strikes me as perhaps the only piece of data that might be helpful to add – analyze the impact of Vta1 loss. On the other hand, if this didn't show something interesting I don’t think it would preclude my enthusiasm for the existing work.

---

## [Author Response]

Reviewer #1:A study of ESCRT protein dynamics on endosomes of live cells is presented. Studies of purified and in vivo overexpressed ESCRT III proteins show that Snf7 and other ESCRT-III subunits are capable of forming spiral filaments that deform membrane; these filaments are proposed to entrap integral membrane proteins and to possess intrinsic membrane bending activity. Prevailing models of ILV biogenesis also propose that the ATPase cycle of Vps4 elicits ESCRT-III filament disassembly and release of ESCRT-III proteins from the membrane, concomitant with vesicle fission. In this study, cutting edge optical and electron microcopy tools were implemented to test several assumptions of the prevailing models by monitoring the association of fluorescently tagged ESCRT-III and Vps4 molecules with yeast endosomes, over time, in live cells. The authors report that ESCRT-III and Vps4 dynamics are inconsistent with current models that posit sequential accumulation of ESCRT-III molecules (i.e., filament growth) that drives ILV membrane budding, concluding with the action of Vps4 to drive ILV fission. Based on their results, a conceptually new mechanistic model is proposed in which continuous association/dissociation of ESCRT-III and Vps4 molecules with the maturing endosome membrane elicits the formation of a dynamic filamentous network that concentrates integral membrane proteins to drive membrane bending via molecular crowding, culminating in vesicle fission. Overall, this is a nicely executed, interesting study that, despite a few technical caveats, is certain to stimulate an important debate within the field.1) Ambiguity regarding the presumed mixed tagged-untagged Snf7 filaments on the endosome is concerning because the degree to which the tag changes Snf7 behavior (i.e., kinetics of filament assembly) is not established. Affected protein functions, if any, might significantly skew the interpretation of the data by restricting analyses to events that do not reflect the physiological situation. In addition, if the assumption that tagged and untagged Snf7 are incorporated with equal probability into an ESCRT-III filament is not correct, then the estimate of Snf7 abundance on the endosome is compromised. I wonder if simple Snf7 fractionation experiments to determine the proportion of tagged Snf7 in membrane-associated cell membrane fraction might lend some clarity to this question and confidence to the authors' assumption.

Although Snf7-eGFP remains stable as a full length fusion protein when precipitated by TCA from total cell lysates (Figure 1—figure supplement 1), it is unfortunately subject to proteolysis during subcellular fractionation; hence it is not possible to carry the quantitative western blot analysis of the membrane fraction suggested by the reviewer.

2) The small proportion of endosomes for which Vps4-mCherry could be detected and analyzed (~30%) begs the question, are the analyzed events representative? Similar to the issue raised in point 1, if the assumption that tagged Vps4 function equivalently to untagged Vps4 (the included tests of tagged Vps4 function are not sufficiently quantitative to determine this), then it is difficult to gauge the significance of restricting the analysis to the subset of endosomes that contain enough Vps4-mCherry to analyze. Have the authors tested if other tags (e.g., halotag) increase Vps4 detection sufficiently to analyze a greater proportion endosomes?

With our current live cell imaging LLSM configuration we can reliable detect 18-20 or more Vps4-mCherry molecules per spot. This is the reason why in the time series ~ 60% of the Snf7-eGFP spots colocalize with Vps4-mCherry. Importantly (an internal control), the remaining non-detected fraction is similar to the fraction of traces with up to 18-20 Vps4-eGFP molecules detected using the live cell imaging LLSM adjusted to detect at least 3 eGFP molecules per spot. These results suggest that Snf7-eGFP always co-localized with Vps4-mCherry.

The full analysis was restricted, however, to 32% of the Snf7-eGFP / Vps4-mCherry traces (still more than 3300) since in these the Vps4-mCherry signal was sufficiently high for reliable tracking during the whole life of the tracing; the relatively fast rate of mCherry bleaching in the remaining traces precluded us from carrying a complete analysis.

We believe that Vps4-eGFP and Vps4-mCherry are fully functional fusion proteins (Figure 1—figure supplement 1). When expressed alone, the lifetimes of Snf7-eGFP, Vps24-eGFP and Vps4-eGFP traces were indistinguishable from each other, consistent with similar recruitment to the endosomes, regardless of being tagged or not.

We have tested Vps4-mScarlet, Vps4-mRuby2, Vps4-turboRFP but Vps4-mCherry still gave the best results. We have not tested the Halo-tag.

Reviewer #2:Teis, Kirchhausen, et al. use a number of new and powerful approaches to provide a detailed view of ESCRT machinery dynamics and function during yeast MVB biogenesis. This work – enabled largely by lattice sheet microscopy – is the first really the first serious look at ESCRT-III and Vps4 dynamics as they generate intralumenal vesicles, and will be of great interest to the field. The data is dense but clearly presented, with balanced interpretations that should help to stimulate new thinking in the field. I recommend publication.1) The authors might clarify the distinction they are making between short and long ESCRT-III polymers; a more or less linear assembly of subunits in ESCRT polymers should allow for either with the observed number of subunits.

The traces shows variable amounts of recruited ESCRT-III; but because this is an ensemble value, we are unable establish whether a given amount corresponds to one or more polymers assembled at the same site.

2) The VPS4EQ data and interpretation is somewhat difficult to follow; the authors should clarify their description of this data and the comparison to the knock-sideways results.

Done.

Reviewer #3:This represents a carefully conducted study that uses quantitative microscopy to address existing models of ESCRT-III assembly, Vps4 recruitment, and ILV formation during MVB sorting. The data supports a conclusion that we need to reconsider existing models pertaining to how these proteins function together and this is valuable as there have been pieces of data in the literature for some time that suggested as much. Data comes from high end instrumentation applied in a manner previously unrealized, is presented in a logical and level manner. While it would appear to be inconsistent with aspects of the existing model it is not clear that it has blown that model out of the water. Nor does it test the new model. Yet there seems little doubt this will drive additional experiments and conversations within the field. I am supportive of publishing this work because I believe the present model is limiting our progress.One thought that comes to mind is that the model puts multiple Vps4 oligomers at the site of ILV formation. A factor that has been implicated in stabilizing interactions between Vps4 oligomers is Vta1. This strikes me as perhaps the only piece of data that might be helpful to add – analyze the impact of Vta1 loss. On the other hand, if this didn't show something interesting I don’t think it would preclude my enthusiasm for the existing work.

Other components, in addition to Vta1, are regulators of Vps4 activity and function. We have generated mutants of Vta1, Did2, Vps60, Ist1 in cells expressing Vps4-eGFP as a way to analyze their function in ESCRT-III and Vps4 dynamics. We plan to use them in the future to further investigate the properties of the ESCRT machinery. We believe, however, that the proposed experiment would go beyond to scope of the current paper.